# From Low Intrinsic Dimensionality to Non-Vacuous Generalization Bounds in Deep Multi-Task Learning

## Abstract

Deep learning methods are known to generalize well from training to future data, even in an overparametrized regime, where they could easily overfit. One explanation for this phenomenon is that even when their *ambient dimensionality*, (i.e. the number of parameters) is large, the models' *intrinsic dimensionality* is small; specifically, their learning takes place in a small subspace of all possible weight configurations.

In this work, we confirm this phenomenon in the setting of *deep multi-task learning*. We introduce a method to parametrize multi-task network directly in the low-dimensional space, facilitated by the use of *random expansions* techniques. We then show that high-accuracy multi-task solutions can be found with much smaller intrinsic dimensionality (fewer free parameters) than what single-task learning requires. Subsequently, we show that the low-dimensional representations in combination with *weight compression* and *PAC-Bayesian* reasoning lead to the first *non-vacuous generalization bounds* for deep multi-task networks.

## 1 Introduction

One of the reasons that deep learning models have been so successful in recent years is that they can achieve low training error in many challenging learning settings, while often also generalizing well from their training data to future inputs. This phenomenon of *generalization despite strong overparametrization* seemingly defies classical results from machine learning theory, which explain generalization by a favorable trade-off between the model class complexity, i.e. how many functions they can represent, and the number of available training examples.

Only recently has machine learning theory started to catch up with practical developments. Instead of overly pessimistic generalization bounds based on traditional complexity measures, such as VC dimension (Vapnik & Chervonenkis, 1971), Rademacher complexity (Bartlett & Mendelson, 2002) or often loose PAC-Bayesian estimates (McAllester, 1998; Alquier, 2024), non-vacuous bounds for neural networks were derived based on *compact encodings* (Zhou et al., 2019; Lotfi et al., 2022). At their core lies the insight that—despite their usually highly overparametrized form—the space of functions learned by deep models has a rather low *intrinsic dimensionality* when they are trained on real-world data (Li et al., 2018). As a consequence, deep models are highly *compressible*, in the sense that a much smaller number of values (or even bits) suffices to describe the learned function compared to what one would obtain by simply counting the number of their parameters.

In this work, we leverage these insights to tackle another phenomenon currently lacking satisfactory generalization theory: the remarkable ability of deep networks for *multi-task learning*. In a setting where multiple related tasks are meant to be learned, high-accuracy deep models that generalize well can be trained from even less training data per task than in the standard single-task setting.

Our contributions are the following: First, we introduce **a new way to parametrize and analyze a class of deep multi-task learning models that allows us to quantify their intrinsic dimensionality.** Specifically, high-dimensional models are learned as linear combinations of a small number of basis elements, which themselves are trained in a low-dimensional random subspace of the original network parametrization.

We then demonstrate experimentally that **for such networks a much smaller effective dimension per task can suffice compared to single-task learning.** Consequently, when learning related tasks, high-accuracy models can be characterized by a very small number of per-task parameters, which suggests an explanation for the strong generalization ability of deep MTL in this setting.

Finally, to formalize the latter observation, **we prove new generalization bounds for deep multi-task and transfer learning based on network compression and PAC-Bayesian reasoning.** The bounds depend only on quantities that are available at training time and can therefore be evaluated numerically. By applying the bounds for models generated by our method and the introduced parameterization, we find that their values are numerically small, making our results **the first non-vacuous bounds for deep multi-task learning**, thereby providing numeric guarantees of the generalization abilities of Deep MTL, not just conceptual ones as in prior work.

## 2 BACKGROUND

**Notation**   For an input set $\mathcal{X}$ and an output set $\mathcal{Y}$, we formalize the concept of a *learning task* as a tuple $t = (p, S, \ell)$, where $p$ is a probability (data) distribution over $\mathcal{X} \times \mathcal{Y}$, $S$ is a dataset of size $m$ that is sampled i.i.d. from $p$, and $\ell : \mathcal{Y} \times \mathcal{Y} \to [0, 1]$ is a loss function. A learning algorithm has access to $S$ and $\ell$, but not $p$, and its goal is to learn a prediction model $f : \mathcal{X} \to \mathcal{Y}$ with as small as possible *risk* on future data, $\mathcal{R}(f) = \mathbb{E}_{(x,y)\sim p} \ell(y, f(x))$.

In a *multi-task setting*, several learning tasks, $t_1, \ldots, t_n$, are meant to be solved. A *multi-task learning (MTL) algorithm* has access to all training sets, $S_1, \ldots, S_n$ and it is meant to output one model per task, $f_1, \ldots, f_n$[1]. The expectation is that if the tasks are related to each other, a smaller overall risk $\mathcal{R}(f_1, \ldots, f_n) = \frac{1}{n} \sum_{i=1}^{n} \mathcal{R}(f_i)$ can be achieved than if each task is learned separately (Caruana, 1997; Thrun & Pratt, 1998; Baxter, 2000).

**Intrinsic dimensionality of neural networks**   Deep networks tend to generalize well to future data, despite the fact that they are typically parametrized with many more weights than the available number of training samples. One explanation suggest to explain this phenomenon is that the set of models that networks actually learn lie on a low-dimensional manifold (Ansuini et al., 2019; Aghajanyan et al., 2021; Pope et al., 2021).

Li et al. (2018) quantified this effect by introducing the *intrinsic dimensionality*: instead of training a model's high-dimensional parameter vector, $\theta \in \mathbb{R}^D$, directly, they represented it indirectly as

$$\theta = \theta_0 + Pw, \tag{1}$$

where $\theta_0$ is a random initialization, and $w \in \mathbb{R}^d$ is a learnable low-dimensional vector, which is expanded to full dimensionality by multiplication with a fixed large matrix, $P \in \mathbb{R}^{D \times d}$, with i.i.d. random entries.

The authors introduced a procedure in which $d$ is determined such that a certain accuracy level is achieved, typically 90% of an ordinarily-trained full-rank model. The resulting *effective dimension* is a property of the model as well as the data, and the authors found that it could usually be chosen orders of magnitude smaller than the *ambient dimension* (number of model parameters) $D$. Later, Aghajanyan et al. (2021) made similar observations in the context of model fine-tuning. Lotfi and co-authors used this observation to establish non-vacuous generalization bounds for learning in the setting of single-task prediction (Lotfi et al., 2022) and generative modeling (Lotfi et al., 2024). To the best of our knowledge, our work is the first that studies and exploits the concept of intrinsic dimensions in the context of multi-task learning.

## 3 AMORTIZED INTRINSIC DIMENSIONALITY FOR MULTI-TASK LEARNING

Our main claim in this work is that the success of deep multi-task learning can be explained by its ability to effectively express and exploit the relatedness between tasks in a low-dimensional subspace of the parametrization space.

---

[1]In practice, one might enforce that these models share certain components, e.g. a token embedding layer or a feature extraction stage. In this work, we do not make any such *a prior* assumptions and let the multi-task learning algorithm decide in what form to share parameters, if any.

To quantify this statement, we extend the definition of intrinsic dimensionality from a single-task setting, where no sharing of information between models takes place, to the multi-task setting, where sharing information between models can occur. Specifically, we introduce the *amortized intrinsic dimension (AID)* that extends the parametrization (1) in a hierarchical way. Instead of learning in a fully random subspace, we divide the multi-task learning task into two components: learning the subspace (for which data of all tasks can be exploited), and learning per-task models within the subspace, for which only the data of the respective task is relevant. Both processes occur simultaneously in an end-to-end fashion, thereby making full use of the high-dimensional model parameter space during the optimization progress.

Formally, the learnable parameters consist of shared vectors $v_1, \ldots, v_k \in \mathbb{R}^l$, and task-specific vectors $\alpha_1, \ldots, \alpha_n \in \mathbb{R}^k$. We use the shared parameters to construct a learned expansion matrix $Q$ as

$$Q = [P_1 v_1, P_2 v_2, \cdots, P_k v_k] \in \mathbb{R}^{D \times k}, \tag{2}$$

where $P_1 \ldots P_k \in \mathbb{R}^{D \times l}$ are fixed matrices with i.i.d. unit Gaussian entries. Models for the individual tasks are learned within the subspace spanned by $Q$, i.e. the parameter vector of the learned model for task $j$ is given by:

$$\theta_j = \theta_0 + Q\alpha_j, \tag{3}$$

again with $\theta_0$ denoting a random initialization.

Overall, this formulation uses $l \cdot k$ parameters to learn an *expansion matrix* that can capture information that is shared across all tasks. In addition, for each of the $n$ tasks, $k$ additional parameters are used to describe a good model within the shared subspace. Consequently, the total number of training parameters in such a parametrization is $lk + nk$, i.e. $\frac{lk}{n} + k$ per task.

The following definition introduces the notions of *intrinsic dimension* and *amortized intrinsic dimension* based on the above construction.

**Definition 1.** For a given model architecture and set of tasks, $t_1, \ldots, t_n$, and a validation accuracy level $\tau$, we define:

- the *(single-task) intrinsic dimensionality*, $\text{ID}_\tau$, as the smallest value for $d$ in the $d$-dimensional expansion (1), such that training the corresponding low-rank models individually on each task results in an average across-tasks accuracy of at least $\tau$.

- the *(multi-task) amortized intrinsic dimensionality*, $\text{AID}_\tau$, as the smallest value for $\frac{lk}{n} + k$ in an $(l, k)$-dimensional expansion (2)-(3), such that multi-task training the corresponding low-rank models results in an average across-tasks accuracy of at least $\tau$.

In order to numerically determine values for the intrinsic dimensionality and the amortized intrinsic dimensionality of Definition 1, one has to select a suitable target accuracy level, $\tau$. In the rest of this work, we do so by adapting the procedure from Li et al. (2018): Assume that well-trained single-task models achieve an average validation accuracy of $A$ across the given tasks. Then we set the target accuracy level as $\tau^* = 0.9A$, and we write $\text{ID}^*$ and $\text{AID}^*$ as shorthand notations for $\text{ID}_{\tau^*}$ and $\text{AID}_{\tau^*}$, respectively. This construction is meant to ensure that, regardless of the tasks characteristics, the target accuracy is not too high to be reachable, but high enough that reaching it does not become trivial.

We now formulate our central scientific hypothesis:

---

**Hypothesis.** *Deep MTL can find models of high accuracy with much smaller* amortized intrinsic dimensionality *than the intrinsic dimensionality required for learning the same tasks separately:*

$$AID^* \ll ID^*$$

.

---

Note that the validity of this hypothesis is not obvious, and its correctness depends on the model architecture as well as set of tasks to be learned. For example, if the tasks are completely unrelated such that no shared subspace is expressive enough to learn good models for all of them, one likely would need one basis element per tasks ($k = n$), and the basis would need the same dimensionality as for single-task learning ($l = \text{ID}^*$). Consequently, $\text{AID}^* = \text{ID}^* + n$, i.e. the multi-task parametrization

Table 1: Intrinsic dimensions for single-task learning ($\text{ID}^*$) and multi-task learning ($\text{AID}^*$) at fixed target accuracies $\text{acc}_{90}$ for different datasets and model architectures (with $D$ parameters, n tasks, and m samples per task) resulting in the reported $\text{ID}^*$ and $\text{AID}^*$.

| Dataset | MNIST SP | MNIST PL | Folktables | Products | split-CIFAR10 | | split-CIFAR100 | |
|---|---|---|---|---|---|---|---|---|
| model | ConvNet | ConvNet | MLP | MLP | ConvNet | ViT | ConvNet | ViT |
| $n/m$ | $30/2000$ | $30/2000$ | $60/900$ | $60/2000$ | $100/453$ | $30/1248$ | $100/450$ | $30/1250$ |
| $D$ | 21840 | 21840 | 11810 | 13730 | 121182 | 5526346 | 128832 | 5543716 |
| $\text{acc}_{90}$ | 85% | 87% | 65% | 75% | 63% | 80% | 40% | 65% |
| $\text{ID}^*$ | 400 | 300 | 50 | 50 | 200 | 200 | 1500 | 550 |
| $\text{AID}^*$ | 31.6 | 166.6 | 10 | 10 | 12 | 26.7 | 36 | 100 |
| $(l, k)$ | (65, 10) | (70, 50) | (60, 5) | (60, 5) | (20, 10) | (50, 10) | (80, 20) | (70, 30) |

is not more parameter-efficient than the single-task one. In the opposite extreme, if all tasks are identical, a single $\text{ID}^*$-dimensional model suffices to represent all solutions, i.e. $k = 1$ and $l = \text{ID}^*$, such that $\text{AID}^* = \frac{\text{ID}^*}{n} + 1$, i.e. the amortized intrinsic dimensionality of multi-task learning shrinks quickly with the number of available tasks. For real-world settings, where tasks are related but not identical, we expect that $k$ and $l$ will have to grow with the number of tasks, but at a sublinear speed compared to $n$.

## 4 DETERMINING THE AID FOR REAL-WORLD TASKS

In this section, we provide evidence for our hypothesis by numerically estimating AID and ID values for different network architectures (fully connected networks, convolutional networks, transformers), different data modalities (tabular, images, text) and different training paradigms (from scratch, fine-tuning). Specifically, we rely on the following six standard multi-task learning benchmarks: *Products*, which consists of vectorial sentence embeddings of text data, and *Folktables*, which contains tabular data, are binary classification tasks, for which we use fully-connected architectures. *MNIST Shuffled Pixels (SP)*, *MNIST Permuted Labels (PL)*, *split-CIFAR10* and *split-CIFAR100* are multi-class image datasets, and we report results for ConvNets and Vision Transformers (ViT). The ViT weight are pretrained on the ImageNet dataset, while the other networks are trained from scratch. For more details on the datasets and network architectures, see Appendix B.

We use a grid search for different values of $l$ and $k$, compute the $\frac{lk}{n} + k$, and choose the smallest value that achieves the desired $\text{acc}_{90}$. Table 1 reports the intrinsic dimensions obtained this way. It clearly supports our hypothesis: in all tested cases, the amortized intrinsic dimensionality in the multi-task setting is smaller than the per-task intrinsic dimensionality when training tasks individually, sometimes dramatically so. For example, to train single-task models on the 30 MNIST SP tasks, it suffices to work in a 400-dimensional subspace of the models' overall 21840-dimensional parameter space. For multi-task learning in representation (3), it even suffices to learn a basis of dimension $k = 10$, where each element is learned within a subspace of dimension $l = 65$ of the model parameters. Consequently, MTL represents all 30 models by just $10 \cdot 65 + 30 \cdot 10 = 950$ parameters, resulting in an amortized intrinsic dimension of just $950/30 \approx 32$. The other datasets show similar trends: for the *Folktables* and *Products* datasets, the intrinsic dimension is reduced from 50 to 10 in both cases. For *split-CIFAR10* and *split-CIFAR100*, the intrinsic dimensions drop from 200 and 1500 to 12 and 36, respectively. The smallest gain we observe is for MNIST PL, where the amortized intrinsic dimensions are slightly more than half of the intrinsic dimension of single-task learning.

From our observations, one can expect that the more tasks are available, the more drastic the difference between single-task ID and multi-task AID can become. Figure 1 visualizes this phenomenon for the MNIST SP dataset. One can indeed see a clear decrease in the AID with a growing number of available tasks, whereas the single-task ID would not be affected by this.

Note that the provided results in Table 1, show the amortized intrinsic dimensionality for our model design. Therefore, the reported values can be seen as upper bounds on the minimum possible amortized intrinsic dimension of multi-task learning. Choosing a specific multi-task design based on prior knowledge about the relatedness of the tasks could push the numbers in Table 1 even lower.

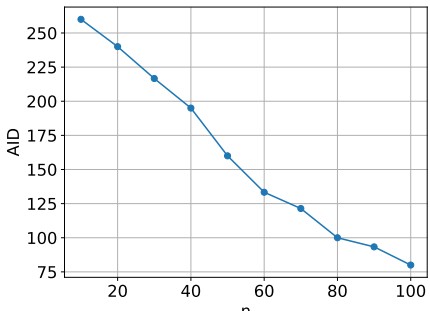 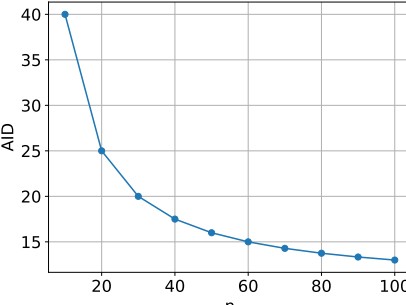

Figure 1: Amortized intrinsic dimensionality, AID*, for multi-task learning on the MNIST PL (left) and MNIST SP (right) datasets ($n = 10, 20, \dots, 100$ tasks, $m = 600$ samples per task). The more tasks are available, the fewer learnable parameters per task are required.

However, even without any such additional information, our reported results clearly confirm our hypothesis by showing a substantial reduction in AID* compared to ID*.

## 5 NON-VACUOUS GENERALIZATION GUARANTEES FOR DEEP MULTI-TASK LEARNING

The result in the previous section suggests that using the representation (3), even quite complex models can be represented with only a quite small number of values. This suggests that the good generalization properties of deep multi-task learning might be explainable by generalization bounds that exploit this fact.

Our main theoretical contribution in this work is a demonstration that this is indeed possible: we derive a new generalization bound for multi-task learning that captures the complexity of all models by their joint encoding length. Technically, we provide high probability upper-bounds for the true tasks $\mathcal{R}(f_1, \dots, f_n)$ based on the training error $\widehat{\mathcal{R}}(f_1, \dots, f_n) = \frac{1}{mn} \sum_{i,j} \ell(y_{i,j}, f_i(x_{i,j}))$, and properties of the model.

### 5.1 BACKGROUND AND RELATED WORK

Before formulating our results, we remind the reader of some classical concepts and results.

**Definition 2.** For any base set, $\mathcal{Z}$, we call a function $E : \mathcal{Z} \to \{0, 1\}^*$ a (prefix-free) *encoding*, if for all $z, z' \in \mathcal{Z}$ with $z \neq z'$, the string $E(z)$ is a not a prefix of the string $E(z')$. The function $l_E : \mathcal{Z} \to \mathbb{N}$, given by $l_E(z) = \text{length}(E(z))$ we call the *encoding length function* of $E$.

Prefix-free codes have a number of desirable properties. In particular, they are uniquely and efficiently decodable. Famous examples include variable-length codes, such as Huffman codes (Huffman, 1952) or Elias codes (Elias, 1975), but also the naive encoding that represents each value of a vector with floating-point values by a fixed-length binary string.

Classical results allow deriving generalization guarantees from the encodings of models.

**Definition 3.** For any set of models, $\mathcal{F}$, an encoder with a base set $\mathcal{Z} = \mathcal{F}$ is called a *model encoder*.

Model encodings can be as simple as storing all model parameters in a fixed length (e.g. 32-bit) floating point format. For a $d$-parameter model, the corresponding length function would simply be $l_E(f) = 32d$. Alternatively, storing only the non-zero coefficient values as (index, value) leads to a length function $l_E(f) = (\lceil \log_2 d \rceil + 32)s$, where $s$ is the number of non-zero parameter values. For highly sparse models, this value can be much smaller than the naive encoding. Over the years, many more involved schemes have been introduced, including, e.g., *weight quantization* (Choi et al., 2020), and *entropy* or *arithmetic coding* of parameter values (Frantar & Alistarh, 2024).

**Theorem 1** (Shalev-Shwartz & Ben-David (2014), Theorem 7.7). *Let $E$ be a model encoding scheme with length function $l_E : \bigcup_{n=1}^{\infty} \mathcal{F}^n \to \mathbb{N}$. Then, for any $\delta > 0$, the following inequality holds with*

*probability at least $1 - \delta$ (over the random training data of size $m$): for all $f \in \mathcal{F}$*

$$\mathcal{R}(f) \leq \widehat{\mathcal{R}}(f) + \sqrt{\frac{l_E(f) \log 2 + \log(\frac{1}{\delta})}{2m}}. \tag{4}$$

Theorem 1 states that models can be expected to generalize well, i.e., have a smaller difference between their training risk and expected risk, if their encodings are short. Specifically, the dominant term in the complexity term of (4) is $\sqrt{\frac{l_E(f) \log 2}{2m}}$. Consequently, for the bound to become non-vacuous (i.e. the right-hand side to be less than 1), in particular, the encoding length must not be much larger than the number of training examples.

### 5.2 COMPRESSION-BASED GUARANTEES FOR MULTI-TASK LEARNING

We now introduce a generalization of Theorem 1 to the multi-task situation.

First, we define two types of encoders, that allow us to formalize the concept of *shared* versus *task-specific* information in multi-task learning.

**Definition 4.** For any model set, $\mathcal{F}$, we define two type of encoders:

- *meta-encoder*: an encoder whose base set is a set of global parameters $E \in \mathcal{E}$ representing shared information between tasks.

- *multi-task (model) encoder*: an encoder with base set $\mathcal{Z} = \bigcup_{n=1}^{\infty} \mathcal{F}^n$ (e.g. arbitrary length tuples of models) that, given a global parameter $E$, encodes a tuple of models with the length $l_E(f_1, \ldots, f_n)$ i.e. encodes the task-specific information.

These encoders generalize (single-task) model encoders in the sense of Definition 3 by allowing more than one model to be encoded at the same time. By exploiting redundancies between the models, shorter encoding lengths can be achievable. For example, if multiple models share certain parts, such as feature extraction layers, those would only have to be encoded once by meta-encoder while multi-task encoder encodes the remaining parts. Additionally, as we will explain in in Section A.4 multi-task encoders can capture unstructured redundancies between task-specific parts to reduce the encoding lengths.

We are now ready to formulate our main results: new generalization bounds for multi-task learning based on the lengths of meta- and multi-task model encoders.

**Theorem 2.** *Let $\mathcal{E}$ be a set of global parameters, and let $l : \mathcal{E} \to \mathbb{N}$ be the length function of the meta-encoder. For each $E \in \mathcal{E}$, let $l_E : \bigcup_{n=1}^{\infty} \mathcal{F}^n \to \mathbb{N}$ be the length function of the multi-task encoder given $E$. For any $\delta > 0$, with probability at least $1 - \delta$ over the sampling of the training data for all $E \in \mathcal{E}$ and for all $f_1, \ldots, f_n \in \mathcal{F}$ the following inequality holds:*

$$\mathcal{R}(f_1, \ldots, f_n) \leq \widehat{\mathcal{R}}(f_1, \ldots, f_n) + \sqrt{\frac{(l(E) + l_E(f_1, \ldots, f_n)) \log(2) + \log \frac{1}{\delta}}{2mn}}. \tag{5}$$

Theorem 2 directly generalizes Theorem 1 to the multi-task situation.

In addition, we also state an alternative *fast-rate bound* based on the PAC-Bayes framework.

**Theorem 3.** *In the settings of Theorem 2, for any multi-task model encoding, any meta-encoding, and any $\delta > 0$, it holds with probability at least $1 - \delta$ over the sampling of the training data:*

$$\mathbf{kl}\left(\widehat{\mathcal{R}}(f_1, \ldots, f_n) | \mathcal{R}(f_1, \ldots, f_n)\right) \leq \frac{(l(E) + l_E(f_1, \ldots, f_n)) \log(2) + \log(\frac{2\sqrt{mn}}{\delta})}{mn}, \tag{6}$$

*where $\mathbf{kl}(q|p) = q \log \frac{q}{p} + (1 - q) \log \frac{1-q}{1-p}$ is the Kullback-Leibler divergence between Bernoulli distributions with mean $q$ and $p$.*

Fast-rate bounds tend to be harder to prove and interpret, but they offer tighter results when the empirical risk is small. Specifically, fast-rate bounds would result in an upper-bound on the true risk by numerically inverting the left-hand side of Theorem 3. This bound scales like $O(\frac{1}{mn})$ when the empirical risk is zero, whereas in Theorem 2 the scaling behaviour is always $O(\frac{1}{\sqrt{mn}})$. For a more detailed discussion and explanation of inverting the bound, see Appendix A.

Table 2: Generalization guarantees (upper bound on test error, *lower is better*) for single-task and multi-task learning. For all tested datasets the multi-task analysis offers better guarantees than the single-task analysis, sometimes by a large margin. The bounds are also always non-vacuous. The fast-rate bound (Theorem 3) offers improved guarantees compared to the more elementary Theorem 2.

| | Dataset
model | MNIST SP
ConvNet | MNIST PL
ConvNet | Folktables
MLP | Products
MLP | split-CIFAR10
ConvNet | ViT | split-CIFAR100
ConvNet | ViT |
|---|---|---|---|---|---|---|---|---|---|
| Single Task | Theorem 1 | 0.612 | 0.576 | 0.566 | 0.332 | 0.874 | 0.660 | 0.994 | 0.906 |
| | $\widehat{\mathcal{R}}$ | 0.229 | 0.194 | 0.272 | 0.160 | 0.310 | 0.182 | 0.637 | 0.376 |
| | $\mathcal{R}$ | 0.239 | 0.205 | 0.279 | 0.160 | 0.406 | 0.203 | 0.714 | 0.417 |
| MTL | (Theorem 2) | 0.230 | 0.404 | 0.394 | 0.222 | 0.529 | 0.319 | 0.869 | 0.665 |
| | (Theorem 3) | 0.196 | 0.350 | 0.388 | 0.203 | 0.527 | 0.280 | 0.830 | 0.658 |
| | $\widehat{\mathcal{R}}$ | 0.101 | 0.066 | 0.272 | 0.139 | 0.305 | 0.106 | 0.627 | 0.274 |
| | $\mathcal{R}$ | 0.096 | 0.064 | 0.268 | 0.141 | 0.331 | 0.114 | 0.637 | 0.313 |

**Discussion.** Theorem 2 establishes that the multi-task generalization gap (the difference between empirical risk and expected risk) can be controlled by a term that expresses how compactly the models can be encoded. Because the inequality is uniform with respect to $E \in \mathcal{E}$, the global parameter can be chosen in a data-dependent way, where tighter guarantees can be achieved if the global parameter itself can be compactly represented.

To get a better intuition of this procedure, we consider two special cases. First, assume a naive encoding, where no global shared parameter is encoded, and the multi-task model encoder simply stores a $D$-dimensional parameter vector for each model in fixed-width form. Then $l(E) = 0$ and $l_E(f_1, \ldots, f_n) = O(nD)$, and the resulting complexity term is of the order $O(\sqrt{\frac{D}{m}})$, as classical VC theory suggests.

In contrast, assume the setting of Section 3, where each task is encoded as a weighted linear combination of $k$ dictionary elements, each of which is computed as the product of a large random matrix with an $l$-dimensional parameter vector. Setting $\mathcal{E}$ to be the set of all such bases, one obtains $l(E) = O(lk)$ (for now disregarding potential additional savings by storing coefficients more effectively than with fixed width bits), and $l_E(f_1, \ldots, f_n) = O(nk)$. Consequently, the complexity term in Theorem 2 becomes $O(\sqrt{\frac{lk/n+k}{m}})$, i.e. the ambient dimension $D$ of the previous paragraph is replaced by exactly the *amortized intrinsic dimension* of Definition 1.

## 5.3 COMPUTING THE BOUNDS NUMERICALLY

Theorem 2 and Theorem 3 are general and can be applied to any multi-task learning methods, however, to achieve non-vacuous results, we need to use models that can achieve both good training performance and low complexity. Therefore, we use models with the parameterization introduced in Section 3. In order to apply our bounds, we use a meta-encoder that encodes the coefficient vectors, $v_1, \ldots, v_k$ of the basis representation (2), and a multi-task model encoder that jointly encodes the collection of per-task coefficients, $\alpha_1, \ldots, \alpha_n$ of the per-task representations 3.

The easiest way would be to allow for arbitrary floating point values in the weights and store them in fixed precision, e.g. 32 bit. The resulting lengths functions would be constant: $l(E) = 32kl$ and $l_E(f_1, \ldots, f_n) = 32nk$, such that the dominant part of the complexity term becomes $O(\sqrt{\frac{\text{AID}}{m}})$ or $O(\frac{\text{AID}}{m})$, respectively. However, it is known that neural networks using quantized weights of fewer bits per parameter can still achieve competitive performance (Han et al., 2016). Therefore, we employ a quantized representation for the parameters we learn, adapting the quantization scheme and learned codebooks of Zhou et al. (2019); Lotfi et al. (2022) to our setting.

Specifically, we use two codebooks, a *global* one with $r_g$ entries, $\mathcal{C}_g = \{c_1, \ldots, c_{r_g}\}$ for quantizing the shared parameters vectors $v_1, \ldots, v_k$, and a *local* one with $r_l$ values, $\mathcal{C}_l = \{c_1, \ldots, c_{r_l}\}$ for quantizing the per-task parameters vectors $\alpha_1, \ldots, \alpha_n$.

Now, for any set of models $f_1, \ldots, f_n$ with global $v_1, \ldots, v_k$, and $\alpha_1, \ldots, \alpha_n$, the *meta-encoder* first stores the $r_g$ codebook entries as 16-bit floating-point values. Then, for each entry of the

Table 3: Generalization guarantees (upper bound on test error, *lower is better*), as well as training errors and test errors for transfer learning versus learning from scratch (no transfer) as averages over 10 runs. For all the datasets, transfer learning results in better guarantees than simply learning a classifier from scratch.

| Dataset model | MNIST SP ConvNet | MNIST PL ConvNet | Folktables MLP | Products MLP | split-CIFAR10 ConvNet | split-CIFAR10 ViT | split-CIFAR100 ConvNet | split-CIFAR100 ViT |
|---|---|---|---|---|---|---|---|---|
| **No transfer** | | | | | | | | |
| Bound | 0.604 | 0.581 | 0.548 | 0.261 | 0.871 | 0.681 | 0.982 | 0.906 |
| $\widehat{\mathcal{R}}$ | 0.220 | 0.199 | 0.256 | 0.101 | 0.304 | 0.206 | 0.628 | 0.374 |
| $\mathcal{R}$ | 0.243 | 0.210 | 0.276 | 0.105 | 0.407 | 0.235 | 0.720 | 0.418 |
| **MTL + Transfer** | | | | | | | | |
| Bound | 0.151 | 0.497 | 0.401 | 0.161 | 0.580 | 0.260 | 0.848 | 0.528 |
| $\widehat{\mathcal{R}}$ | 0.094 | 0.219 | 0.261 | 0.094 | 0.277 | 0.134 | 0.648 | 0.345 |
| $\mathcal{R}$ | 0.088 | 0.218 | 0.268 | 0.101 | 0.304 | 0.135 | 0.664 | 0.333 |

parameter vectors, it computes the index of the codebook entry that represents its value and it stores all indices together using arithmetic coding (Langdon, 1984), which exploits the global statistics of the index values. Analogously, the multi-task model encoder first stores the $l$ codebook entries. It then represents the entries of $\alpha_1, \ldots, \alpha_n$ as indices in the code, and encodes them jointly using arithmetic coding. Note that while the latter process of encoding the model coefficients does not depend on the shared parameters, its decoding process does, because the actual network weights can only be recovered if also the representation basis, $Q$, is known.

In practice, training networks with weights restricted to a quantized set is generally harder than training in ordinary floating-point form. Therefore, for our experiments, we follow the procedure from Lotfi et al. (2022). We first train all networks in unconstrained form. We then compute codebooks by (one-dimensional) $k$-means clustering of the occurring weights, and quantizing each weight to the closest cluster center, and then fine-tune the quantized network.

The resulting guarantees then hold for the quantized networks. For Theorem 2, we can read off the guarantees on the multi-task risk directly from the explicit complexity term and the training error. For Theorem 3, we have to invert the **kl**-expression, which is easily possible numerically. See Appendix A for more details.

The results are reported in Table 2 which compares our standard and fast-rate MTL bounds with the (sota) fast-rate single-task bound of Lotfi et al. (2022), which shows the advantage of MTL over single-task learning for several different datasets and model architectures.

An advantage of our generalization bounds is that they allow us to encode all tasks together. Formally, the bound is based on the term $l_E(f_1, \ldots, f_n)$ and not the naive $\sum_{i=1}^n l_E(f_i)$. Given the conceptual connections between compressibility, information, and entropy, the difference between these two quantities can be seen as a computable approximation to the mutual information between the tasks Li & Vitányi (2019). For more details see Appendix A.4.

## 5.4 APPLICATION TO TRANSFER LEARNING

The multi-task setting of Section 3 has a straightforward extension to the *transfer learning* setting [2]. Assume that, after multi-task learning as in previous sections, we are interested in training a model for another related task. Then, a promising approach is to do so in the already-learned representation, given by the learned matrix $Q$. In particular, this procedure has a low risk of overfitting, because $Q$ is low-dimensional and has been constructed without any training data for the new task. In terms of formal guarantees, $Q$ is a fixed quantity, so no complexity terms appear for it in a generalization bound. However, learning purely in this representation would pose the risk of underfitting, because the MTL representation might not actually be expressive enough also for new tasks.

Consequently, we suggest a transfer learning method with few parameters that combines the flexibility of single-task learning with the advantages of a learned representation. Specifically, for any new task,

---

[2]Note that our transfer learning setup differs from the one in Lotfi et al. (2022): to transfer information from previous tasks to new ones we use MTL to identify an extremely low-dimensional basis, in which the subsequent learning of new tasks takes place. In contrast, they use a random basis, and information from a previous task enters as an offset, which corresponds to our experiments with a pre-trained network in Section 4.

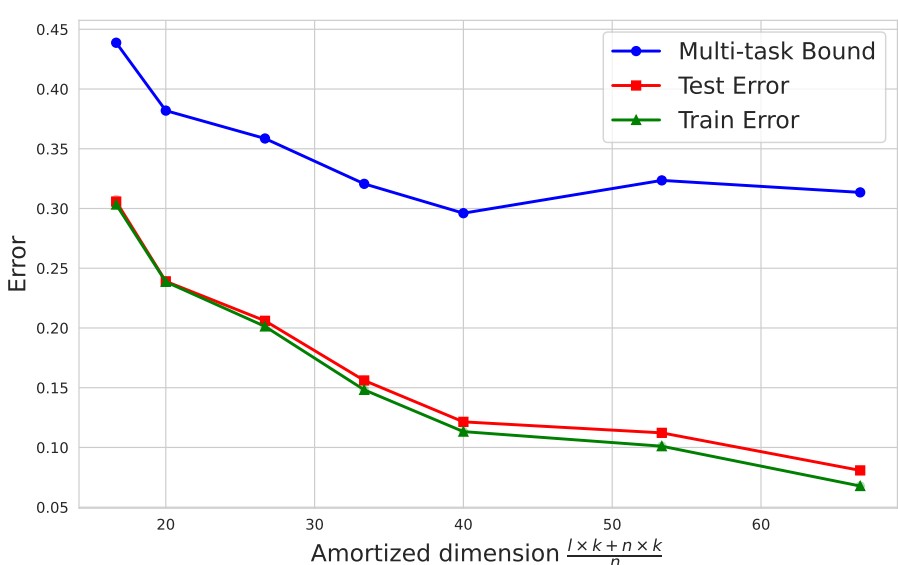

Figure 2: Comparison of Test Error and MTL upper bound.

we learn a model in a subspace of $d' = k + k'$ dimensions, of which $k$ basis vectors stems from the representation matrix $Q$ (2), and the remaining $k'$ come from a random basis as in (1), $k'$ is a tunable hyperparameter. Formally, the new model has the following representation:

$$\theta_j = \theta_0 + Q\alpha + Pw, \tag{7}$$

in which $Q \in \mathbb{R}^{D \times k}$ is the result of multi-task learning, $P \in \mathbb{R}^{D \times k'}$ is a random matrix, and $\alpha \in \mathbb{R}^k$ and $w \in \mathbb{R}^{k'}$ are learnable parameters for the new task. If the previous tasks are related to the current task, we would expect to have a smaller dimension $d'$ compared to the case that we do not use $Q$.

Theorem 1 yields generalization guarantees for this situation, in which only the encoding length of $\alpha$ and $w$ enters the complexity term, while $Q$ (like $P$) do not enter the bound. To encode the coefficients $\alpha$ and $w$, we can reuse the codebook of the MTL situation, which is now fixed and therefore does not have to be encoded or learn (and encode) a new one. Table 3 shows the resulting generalization guarantees. For all datasets, learning a model using the pre-trained representation leads to stronger guarantees than single-task training from scratch. The bounds are clearly non-vacuous, despite the fact that only a single rather small training set is available in each case.

## 5.5 COMPARISON OF TEST ERROR AND MTL UPPER BOUND.

In Figure 2 we plot the values of the actual test error and the computed upper bounds for different values of $l, k$, for the ViT experiment on the split-CIFAR10 dataset. As it is shown in the figure, both the values of the bound and the test error decrease with more parameters and increased expressiveness. However, the bound reaches a point where the cost of the addition of expressiveness exceeds the benefit from having a smaller training error. In contrast, the actual test error would not increase, as it is observed empirically for neural networks, and more progress is needed to make the bounds tighter.

## 6 RELATED WORK

*Multi-task learning* (Baxter, 1995; Caruana, 1997) has been an active area of research for many years, first in classical (shallow) machine learning and then also in deep learning, see, e.g., Yu et al. (2024) for a recent survey. Besides practical methods that typically concentrate on the question of how to share information between related tasks (Kang et al., 2011; Sun et al., 2020) and how to establish

their relatedness (Juba, 2006; Daumé III & Kumar, 2013; Fifty et al., 2021), there has also been interest from early on to theoretically understand the generalization properties of MTL.

A seminal work in this area is Maurer (2006), where the author studies the feature learning formulation of MTL in which the systems learn a shared representation space and individual per-task classifiers within that representation. In the case of linear features and linear classifiers, he derives a generalization bounds based on Rademacher-complexity. Subsequently, a number of follow-up works extended and refined the results to cover other regularization schemes or complexity measures (Crammer & Mansour, 2012; Pontil & Maurer, 2013; Yousefi et al., 2018; Pentina & Lampert, 2017; Du et al., 2021). However, none of these results are able to provide non-vacuous bounds for overparametrized models, such as deep networks.

A related branch of research targets the problems of representation or meta-learning. There, also multiple tasks are available for training, but the actual problem is to provide generalization guarantees for future learning tasks (Baxter, 2000) A number of works in this area rely on PAC-Bayesian generalization bounds (Pentina & Lampert, 2014; Amit & Meir, 2018; Liu et al., 2021; Guan & Lu, 2022; Riou et al., 2023; Friedman & Meir, 2023; Rezazadeh, 2022; Rothfuss et al., 2023; Ding et al., 2021b; Tian & Yu, 2023; Farid & Majumdar, 2021; Zakerinia et al., 2024), as we do for our Theorem 3. This makes them applicable to arbitrary model classes, including deep networks. These results, however, hold only under specific assumptions on the observed tasks, typically that these are themselves i.i.d. sample from a task environment. The complexity terms in their bounds are either not numerically computable, or vacuous by several orders of magnitude for deep networks, because they are computed in the ambient dimensions, not the intrinsic ones. Another difference to our work is that, even those parts of their bound that relate to multi-task generalization are sums of per-task contributions, which prevents the kind of synergetic effects that our joint encoding can provide.

## 7 CONCLUSION

We presented a new parametrization for deep multi-task learning problems that directly allows one to read off its intrinsic dimension. We showed experimentally that for common benchmark tasks, the amortized intrinsic dimension per task can be much smaller than when training the same tasks separately. We derived two new encoding-based generalization bounds for multi-task learning, that result in non-vacuous guarantees on the multi-task risk for several standard datasets and model architectures. To our knowledge, this makes them the first non-vacuous bounds for deep multi-task learning.

A limitation of our analysis in Section 3 is that it relies on a specific representation of the learned models. As such, the values we obtain for the amortized intrinsic dimensionality constitute only upper bounds to the actual, i.e. minimal, number of necessary degrees of freedom. In future work, we plan to explore the option of extending our results also to other popular multi-task parametrization, such as methods based on prototypes, MAML (Finn et al., 2017), or feature learning (Collobert & Weston, 2008). Note that our theoretical results in Section 5 readily apply to such settings, as long as we define suitable encoders. In this context, as well as in general, it would be interesting to study if our bounds could be improved further by exploiting more advanced techniques for post-training weight quantization (Rastegari et al., 2016; Frantar et al., 2023) or learning models directly in a quantized form (Hubara et al., 2018; Wang et al., 2023). Another promising direction for future work would be to scale our results to even larger datasets and large language models, which are currently out of the scope of this paper.

## 8 REPRODUCIBILITY STATEMENT

The proof of all theoretical results is included in Appendix A. The code for reproducing the experiments is submitted as supplementary material, and the experimental details are included in Appendix B.

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

# A  PROOFS

## A.1  COMPARISON OF THE BOUNDS

Theorem 3 which in the PAC-Bayes literature is referred to as a *fast-rate* bound, can be tighter than the bound of Theorem 2. Specifically, up to a different in the $\log$-term we can obtain the bound of Theorem 2 by relaxing the bound of Theorem 3. Based on Pinsker's inequality, we have $2(p-q)^2 \leq \mathbf{kl}(q|p)$ and therefore

$$2\Big(\widehat{\mathcal{R}}(f_1,\ldots,f_n) - \mathcal{R}(f_1,\ldots,f_n)\Big)^2 \leq \mathbf{kl}(\widehat{\mathcal{R}}(f_1,\ldots,f_n)|\mathcal{R}(f_1,\ldots,f_n)). \tag{8}$$

Combining Theorem 3 and equation (8) gives that for every $\delta > 0$ with probability at least $1 - \delta$, we have:

$$\mathcal{R}(f_1,...,f_n) \leq \widehat{\mathcal{R}}(f_1,...,f_n) + \sqrt{\frac{(l(E) + l_E(f_1,...,f_n))\log(2) + \log\frac{2\sqrt{mn}}{\delta}}{2mn}}. \tag{9}$$

This is very similar to the bound of Theorem 2, differing by the additional term $\frac{\log 2\sqrt{mn}}{2mn}$, which is negligible for large $m$ or $n$.

Alternatively, instead of using Pinsker's inequality, we can numerically find a better upper-bound for $\mathcal{R}(f_1,\ldots,f_n)$. Following Seeger (2002); Alquier (2024), we define

$$kl^{-1}(q|b) = \sup\{p \in [0,1] : \mathbf{kl}(q|p) \leq b\}. \tag{10}$$

**Corollary 4.** *With the same setting as Theorem 3, with probability at least $1 - \delta$, we have:*

$$\mathcal{R}(f_1,\ldots,f_n) \leq kl^{-1}\Bigg(\widehat{\mathcal{R}}(f_1,\ldots,f_n))\bigg|\frac{(l(E) + l_E(f_1,\ldots,f_n))\log(2) + \log(\frac{2\sqrt{mn}}{\delta})}{mn}\Bigg). \tag{11}$$

Note that for a fixed $q \in (0,1)$, the function $\mathbf{kl}(q|p)$ is minimized for $p = q$, and it is a convex increasing function in $p$, when $q \leq p \leq 1$. Therefore, we can find an upper bound for $kl^{-1}(q|b)$, by binary search in the range $[q,1]$, or Newton's method as in (Dziugaite & Roy, 2017).

As Table 2 shows, the numeric bounds obtained this way can be tighter bound than the upper bound from the Theorem 2.

## A.2  PROOF OF THEOREM 2

**Theorem 2.** *Let $\mathcal{E}$ be a set of global parameters, and let $l : \mathcal{E} \to \mathbb{N}$ be the length function of the meta-encoder. For each $E \in \mathcal{E}$, let $l_E : \bigcup_{n=1}^{\infty} \mathcal{F}^n \to \mathbb{N}$ be the length function of the multi-task encoder given $E$. For any $\delta > 0$, with probability at least $1 - \delta$ over the sampling of the training data for all $E \in \mathcal{E}$ and for all $f_1,...,f_n \in \mathcal{F}$ the following inequality holds:*

$$\mathcal{R}(f_1,...,f_n) \leq \widehat{\mathcal{R}}(f_1,...,f_n) + \sqrt{\frac{(l(E) + l_E(f_1,...,f_n))\log(2) + \log\frac{1}{\delta}}{2mn}}. \tag{5}$$

*Proof.* We rely on standard arguments for coding-based generalization bounds, which we adapt to the setting of multiple tasks with potentially different data distributions.

For any $i = 1,\ldots,n$, let $S_i = \{z_{i,1},\ldots,z_{i,m}\} \subset \mathcal{Z}$ be the training data available for a task $t_i$ and let $\ell_i : \mathcal{F} \times \mathcal{Z}$ be its loss function. For any tuple of models, $F = (f_1,\ldots,f_n)$, we define random variables $X_{i,j} = \frac{1}{mn}\ell_i(f_i, z_{i,j}) \in [0, \frac{1}{mn}]$, where $n$ is the number of tasks and $m$ is the number of samples per task, such that

$$\widehat{\mathcal{R}}(F) := \widehat{\mathcal{R}}(f_1,\ldots,f_n) = \sum_{(i,j)\in I} X_{i,j}, \tag{12}$$

for $I = \{(i,j) :\ i \in \{1,\ldots,n\} \wedge j \in \{1,\ldots,m\}\}$, and

$$\mathcal{R}(F) := \mathcal{R}(f_1,\ldots,f_n) = \mathbb{E}\Big[\sum_{(i,j)} X_{i,j}\Big]. \tag{13}$$

Therefore, based on Hoeffding's inequality over these random variables, we have for any $t \geq 0$:

$$\mathbb{P}\left\{\mathcal{R}(F) \geq \widehat{\mathcal{R}}(F) + t\right\} \leq e^{-2t^2 mn}. \tag{14}$$

Now, assume a fixed $\delta > 0$. For any meta-encoder, $E$, and any tuple of models, $F = (f_1, ..., f_n)$, we define a weight $w_{E;F} = \delta \cdot 2^{-l(E)} 2^{-l_E(f_1, ..., f_n)}$, where $l(\cdot)$ is the length function of the meta-encoder and $l_E(\cdot)$ is the length function of the encoder $E$. We instantiate (14) with a value $t_{E;F}$ such that $e^{-t_{E;F}^2 mn} = w_{E;F}$, i.e. $t_{E;F} = \sqrt{-\frac{\log w_{E;F}}{2mn}}$. Taking a union bound over all tuples $(E, F)$ and observing that $\sum_{E;F} w_{E;F} = \delta \cdot \sum_E 2^{-l(E)}\left(\sum_F 2^{-l_E(F)}\right) \leq \delta$, because of the Kraft-McMillan inequality for prefix codes (Kraft, 1949; McMillan, 1956), we obtain

$$\mathbb{P}\left\{\exists E, F : \mathcal{R}(F) - \widehat{\mathcal{R}}(F) \geq \sqrt{\frac{\left(l(E) + l_E(F)\right)\log(2) + \log\frac{1}{\delta}}{2mn}}\right\} \leq \delta. \tag{15}$$

By rearranging the terms, we obtain the claim of Theorem 2. □

### A.3 PROOF OF THEOREM 3.

In this section, we provide the proof for Theorem 3. Since our hypothesis set is discrete, we use a union-bound approach, similar to the proof of Theorem 2. A similar result can be proved using the PAC-Bayes framework and change of measure, similar to the fast-rate bounds in Guan & Lu (2022), but we found our version to be conceptually simpler.

**Theorem 3.** *In the settings of Theorem 2, for any multi-task model encoding, any meta-encoding, and any $\delta > 0$, it holds with probability at least $1 - \delta$ over the sampling of the training data:*

$$\mathbf{kl}\left(\widehat{\mathcal{R}}(f_1, \ldots, f_n) | \mathcal{R}(f_1, \ldots, f_n)\right) \leq \frac{(l(E) + l_E(f_1, \ldots, f_n))\log(2) + \log(\frac{2\sqrt{mn}}{\delta})}{mn}, \tag{6}$$

*where $\mathbf{kl}(q|p) = q\log\frac{q}{p} + (1 - q)\log\frac{1-q}{1-p}$ is the Kullback-Leibler divergence between Bernoulli distributions with mean $q$ and $p$.*

*Proof.* For any fixed tuple of models, $F = (f_1, \ldots, f_n)$, we have

$$\mathbb{P}(\mathbf{kl}(\widehat{\mathcal{R}}(F)|\mathcal{R}(F)) \geq t) = \mathbb{P}(e^{mn\,\mathbf{kl}(\widehat{\mathcal{R}}(f)|\mathcal{R}(f))} \geq e^{mnt}) \tag{16}$$

$$\leq \frac{\mathbb{E}[e^{mn\,\mathbf{kl}(\widehat{\mathcal{R}}(f)|\mathcal{R}(f))}]}{e^{mnt}} \leq \frac{2\sqrt{mn}}{e^{mnt}}. \quad \text{(Lemma 7 below)} \tag{17}$$

Subsequently, we follow the steps of the proof of Theorem 2. Assume a fixed $\delta > 0$. For any meta-encoder, $E$, and any tuple of models, $F = (f_1, ..., f_n)$, we define a weight $w_{E;F} = \delta \cdot 2^{-l(E)} 2^{-l_E(f_1, ..., f_n)}$, where $l(\cdot)$ is the length function of the meta-encoder and $l_E(\cdot)$ is the length function of the encoder $E$. We instantiate (17) with a value $t_{E;F}$ such that $2\sqrt{mn}e^{-t_{E;F} mn} = w_{E;F}$, i.e. $t_{E;F} = -\frac{\log w_{E;F}}{mn}$. Therefore, we have with probability at least $1 - w_{E,F}$:

$$\mathbf{kl}(\widehat{\mathcal{R}}(f)|\mathcal{R}(f)) \leq \frac{l(E) + l_E(f_1, \ldots, f_n))\log(2) + \log\frac{2\sqrt{mn}}{\delta}}{mn}. \tag{18}$$

Taking a union bound over all tuples $(E, F)$ and observing that $\sum_{E;F} w_{E;F} = \delta \cdot \sum_E 2^{-l(E)}\left(\sum_F 2^{-l_E(F)}\right) \leq \delta$, because of the Kraft-McMillan inequality for prefix codes (Kraft, 1949; McMillan, 1956), we obtain

$$\mathbb{P}\left\{\forall E, f_1, \ldots, f_n : \mathbf{kl}(\widehat{\mathcal{R}}(f)|\mathcal{R}(f)) \leq \frac{l(E) + l_E(f_1, \ldots, f_n))\log(2) + \log\frac{2\sqrt{mn}}{\delta}}{mn}\right\} \leq \delta. \tag{19}$$

□

## A.4 DISCUSSION

**Encoding all tasks together**  An advantage of our generalization bounds is that they allow us to encode all tasks together. Formally, the bound is based on the term $l_E(f_1, \ldots, f_n)$ and not the naive $\sum_{i=1}^n l_E(f_i)$. Given the conceptual connections between compressibility, information, and entropy, the difference between these two quantities can be seen as a computable approximation to the mutual information between the tasks Li & Vitányi (2019).

In practice, the ability to encode tasks jointly helps in particular when using arithmetic coding, which can exploit redundancies between tasks representations on different levels. Specifically, to encode the indices of relevant codebook entries, one encodes the length of the codebook, the empirical fraction of occurrences of the indices, and the arithmetic coding given the fractions. For example, when performing multi-task learning on the Folktables dataset, in which tasks are highly related, each of the 60 models was characterized as a 5-dimensional vector ($k = 5$), resulting in 300 task-specific parameters overall. To encode these 300 parameters, we used a codebook with $r_l = 10$, which required 160 bits for the codebook. Encoding the empirical fractions required 90 bits, and arithmetic coding additional 874 bits. Overall, the encoding length was 1124 bits in total, or $\approx 18.7$ bits per task. In contrast, encoding the same task-specific models separately would result in the sum of encoding equal to 2591 bits or 43.2 bits per task. This is actually more than double compared to the size of the joint encoding.

**Comparison to the fast-rate bounds of Guan & Lu (2022):**  As mentioned earlier, Guan & Lu (2022) proved a fast-rate bound for meta-learning which consists of a multi-task bound and a meta bound. The multi-task bound provided here, shares a similar structure with the bound of Guan & Lu (2022), and a similar approach to upper-bound the MGF (Moment generating function). There are three main differences in how to use the bounds. The first one is that they approximate the upper-bound for the $\mathcal{R}(F)$ based on a closed-form approximation, which has poorer performance compared to the numerical optimization, and even in the cases where the empirical error is not small, their approximation can be even worse than the bound of Theorem 2). The second difference is that they use Gaussian distributions on the networks that scale with ambient dimensionality, making the bounds vacuous by several orders of magnitude. The more structural difference is that the sum of the task-specific complexity terms appears in their bound. On the other hand, we have a joint complexity term for the task-specific part (which in our case is the length of the multi-task encoding), which as explained above is much smaller than the sum of individual ones.

**Task-relatedness:**  To check the effect of task relatedness in our results, we report an experiment with Shuffle Pixel data in which a different number of pixels can change. In this experiment, we shuffle 100 pixels instead of 200 pixels as in the Tables 1 and 2, therefore, the tasks are more related. This results in to decrease in the AID from 31.6 to 28.3, and an improvement in the bounds from 0.23 to 0.21, which is consistent with tasks being more related.

## A.5 AUXILIARY LEMMAS

**Lemma 5.** *(Berend & Tassa, 2010)[Proposition 3.2] Let $X_i$, $1 \leq i \leq t$, be a sequence of independent random variables for which $P(0 \leq X_i \leq 1) = 1$, $X = \sum_{i=1}^t X_i$, and $\mu = E(X)$. Let $Y$ be the binomial random variable with distribution $Y \sim B\left(t, \frac{\mu}{t}\right)$. Then for any convex function $f$ we have:*

$$\mathbf{E}f(X) \leq \mathbf{E}f(Y). \tag{20}$$

**Lemma 6.** *(Maurer, 2004)[Theorem 1] Let $Y$ be the binomial random variable with distribution $Y \sim B\left(t, \frac{\mu}{t}\right)$. Then we have:*

$$\mathbb{E}[e^{t \; \mathbf{kl}(\frac{Y}{t}|\frac{\mu}{t})}] \leq 2\sqrt{t}, \tag{21}$$

*where $\mathbf{kl}(q|p) = q \log \frac{q}{p} + (1 - q) \log \frac{1-q}{1-p}$ is the Kullback-Leibler divergence between Bernoulli distributions with mean $q$ and $p$.*

**Lemma 7.** *Given the $n$ datasets $S_1, \ldots, S_n$ of size $m$. For fixed models, $F = (f_1, \ldots, f_n)$, we have*

$$\mathbb{E}[e^{mn \; \mathbf{kl}(\widehat{\mathcal{R}}(F)|\mathcal{R}(F))}] \leq 2\sqrt{mn}. \tag{22}$$

*Proof.* Let $f$ be the function $f(x) = mn\,\mathbf{kl}(\frac{x}{mn}|\mathcal{R}(F))$, this function is convex. If we define $X_{i,j} = \ell_i(f_i, z_{i,k})$, since $f_i$s are fixed, and samples are independent, the random variables $X_{i,j}$s are independent. Therefore, $\sum X_{i,j} = mn\widehat{\mathcal{R}}(F)$, and $\mathbb{E}[\sum X_{i,j}] = mn\mathcal{R}(F)$. Let $Y \sim B(mn, \mathcal{R}(F))$. Because of Lemma 5 we have

$$\mathbb{E}[e^{f(mn\widehat{\mathcal{R}}(F))}] \leq \mathbb{E}[e^{f(Y)}], \tag{23}$$

or equivalently,

$$\mathbb{E}[e^{(mn\,\mathbf{kl}(\widehat{\mathcal{R}}(F)|\mathcal{R}(F))}] \leq \mathbb{E}[e^{mn\,\mathbf{kl}(\frac{Y}{mn}|\mathcal{R}(F))}]. \tag{24}$$

Because of Lemma 6, we have:

$$\mathbb{E}[e^{mn\,\mathbf{kl}(\frac{Y}{mn}|\mathcal{R}(F))} \leq 2\sqrt{mn}. \tag{25}$$

Combining these two inequalities completes the proof. □

## B  EXPERIMENTAL DETAILS

In this section, we provide the details of our experiments. Code for reproducing the experiments is included in the supplementary materials.

### B.1  DATASETS

We use six standard datasets that have occurred in the theoretical multi-task learning literature before.

**MNIST Shuffled Pixels (SP):** (Amit & Meir, 2018) each task is a random subset of the MNIST (LeCun & Cortes, 1998) dataset in which 200 of the input pixels are randomly shuffled. The same shuffling is consistent across all samples of that task.

**MNIST Permuted Labels (PL):** (Amit & Meir, 2018) like MNIST-SP, but instead of shuffling pixels, the label ids of each task are randomly (but consistently) permuted.

**Folktables:** (Ding et al., 2021a) A tabular dataset consisting of public US census information. From personal features, represented in a binary encoding, the model should predict if a person's income is above or below a threshold. Tasks correspond to different geographic regions.

**Multi-task dataset of product reviews (MTPR):** (Pentina & Lampert, 2017) The data points are natural language product reviews, represented as vectorial sentence embeddings. The task is to predict if the sentiment of the review is positive or negative. Each product forms a different task.

**split-CIFAR10:** (Zhao et al., 2018) tasks are created by randomly choosing a subset of 3 labels from the CIFAR10 dataset (Krizhevsky, 2009) and then sampling images corresponding to these classes.

**split-CIFAR100:** (Zhao et al., 2018) like split-CIFAR10, but using label subsets of size 10 and images from the CIFAR100 dataset (Krizhevsky, 2009).

### B.2  MODEL ARCHITECTURES

For the MNIST experiments, we use convolutional networks used in Amit & Meir (2018). For the vectorial dataset *Product* and the tabular dataset *Folktables*, we use 4-layer fully connected networks. For the CIFAR experiments, we use two different networks: 1) The CNN used in Scott et al. (2024), and a ViT model pretrained from ImageNet (Dosovitskiy et al., 2021). The details of the model architectures are provided in Table 6.

The models' ambient dimensions (number of network weights) range from approximately 12000 to approximately 5.5 million, i.e. far more than the available number of samples per task. As random matrices $P$ for the single-task parametrization (1), we use the Kronecker product projector of Lotfi et al. (2022), $P = Q_1 \otimes Q_2/\sqrt{D}$, for $Q_1, Q_2 \sim \mathcal{N}(0,1)^{\sqrt{D} \times \sqrt{d}}$. By this construction, the matrix $P \in \mathbb{R}^{D \times d}$ never has to be explicitly instantiated, which makes the memory and computational overhead tractable. For the multi-task representation (3), we use the analogous construction to form $Q' = [P_1, P_2, \ldots, P_k] = Q'_1 \otimes Q'_2/\sqrt{D} \in \mathbb{R}^{D \times kl}$.

In this section, we provide experimental details to reproduce the results.

### B.3 MODEL TRAINING DETAILS

All models are implemented in the PyTorch framework. We train them for 400 epochs with Adam optimizer, weight decay of 0.0005, and learning rate from $\{0.1, 0.01, 0.001\}$. The hyperparameter $l$ (the dimensionality of the random matrices which build $Q$) is chosen from values in $\{20, 30, 40, 50, 60, 70, 80, 90, 100, 120, 150, 200, 300, 400, 500, 600, 700, 800, 900, 1000, 1200, 1400, 1600, 1800, 2000, 2500, 3000, 3500, 4000, 5000, 6000, 7000, 8000\}$. The hyperparameter $k$ (the number of basis vectors in $Q$) is chosen from values in $\{5, 10, 15, 20, 30, 35, 40, 50, 60, 70, 80, 90\}$.

We train all shared and task-specific parameters jointly. After the training is over, we first quantize the shared parameters, and after fixing them, we quantize each task-specific parameter separately. Quantization training is done with 30 epochs using SGD with a learning rate of $0.0001$. For the codebook size, for the single-task learning, we choose it from $\{2, 3, 5, 10, 15, 20, 30, 40\}$. For multi-task we choose the codebook size for shared parameters ($r_G$) from $\{10, 15, 20, 30\}$ and for encoding the joint task-specific vectors we use codebook size ($r_l$) from $\{3, 10, 15, 20, 25, 30\}$. For transfer learning, we use a 1-bit hyperparameter to decide if we want to learn a small new codebook for the new task or transfer the codebook from the multi-task learning stage. For each dataset, the hyperparameters are also encoded and considering in compute the length encoding, since we tune the hyperparameter in a data-dependent way.

The detailed numeric values used for quantizing parameters in Table 2 are shown in Table 4 and Table 5.

Table 4: Numeric values contributing to the generalization bounds in Table 2 for $n$: number of tasks, $m$: average number of examples per task, $L$: number of classes.

|  | dataset | MNIST SP | MNIST PL | Folktables | Products |
|---|---|---|---|---|---|
|  | $n\,/\,m\,/\,L$ | $30\,/\,2000\,/\,10$ | $30\,/\,2000\,/\,10$ | $60\,/\,900\,/\,2$ | $60\,/\,2000\,/\,2$ |
| Single Task | training error | 0.229 | 0.194 | 0.272 | 0.160 |
|  | test error | 0.239 | 0.205 | 0.279 | 0.160 |
|  | Upper bound on $\mathcal{R}$ | 0.612 | 0.576 | 0.566 | 0.332 |
|  | codebook size $r$ | 10 | 10 | 5 | 5 |
|  | (average) encoding length | 854.0 | 862.6 | 211.7 | 216.4 |
| Multi-task | training error | 0.101 | 0.066 | 0.272 | 0.139 |
|  | test error | 0.096 | 0.064 | 0.268 | 0.141 |
|  | Upper bound on $\mathcal{R}$ | 0.196 | 0.350 | 0.388 | 0.203 |
|  | codebook sizes $r_g\,/\,r_l$ | 10 / 3 | 15 / 10 | 10 / 10 | 10 / 10 |
|  | $l(E)$ | 2323 | 14887 | 1586 | 1192 |
|  | $l_E(f_1, \ldots, f_n)$ | 508 | 4796 | 686 | 1128 |
|  | (average) encoding length | 94.4 | 651.1 | 37.9 | 38.7 |

Table 5: Numeric values contributing to the generalization bounds in Table 2.

| dataset | | split-CIFAR10 | | split-CIFAR100 | |
|---|---|---|---|---|---|
| Model | | CNN | ViT | CNN | ViT |
| $n\,/\,m\,/\,L$ | | $100\,/\,453\,/\,3$ | $30\,/\,1248\,/\,3$ | $100\,/\,450\,/\,10$ | $30\,/\,1250\,/\,10$ |
| | training error | 0.310 | 0.182 | 0.637 | 0.376 |
| | test error | 0.406 | 0.203 | 0.714 | 0.417 |
| Single Task | Upper bound on $\mathcal{R}$ | 0.874 | 0.660 | 0.994 | 0.906 |
| | codebook size $r$ | 10 | 10 | 10 | 10 |
| | (average) encoding length | 544.4 | 860.8 | 542.0 | 1529.2 |
| | training error | 0.305 | 0.106 | 0.627 | 0.274 |
| | test error | 0.331 | 0.114 | 0.637 | 0.313 |
| | Upper bound on $\mathcal{R}$ | 0.527 | 0.280 | 0.830 | 0.658 |
| Multi-task | codebook sizes $r_g\,/\,r_l$ | 10 / 20 | 10 / 20 | 20 / 10 | 20 / 30 |
| | $l(E)$ | 2358 | 3109 | 4209 | 11512 |
| | $l_E(f_1, \ldots, f_n)$ | 4144 | 1747 | 3341 | 4957 |
| | (average) encoding length | 65.0 | 161.9 | 75.5 | 549.0 |

Table 6: Model Architectures

| Datasets | Layer | Details |
|---|---|---|
| MNIST-SP/ MNIST-PL | Conv1 | Conv2d(input: $C$, output: 10, kernel: $5 \times 5$) |
| | Activation | ELU |
| | Pooling | MaxPool2d(kernel: $2 \times 2$) |
| | Conv2 | Conv2d(input: 10, output: 20, kernel: $5 \times 5$) |
| | Activation | ELU |
| | Pooling | MaxPool2d(kernel: $2 \times 2$) |
| | Flatten | - |
| | FC1 | Linear(input: Conv Output, output: 50) |
| | FC_out | Linear(input: 50, output: Output_dim) |
| Products/ Folktables | FC1 | Linear(input: input_dim, output: 128) |
| | Activation | ReLU |
| | FC2 | Linear(input: 128, output: 64) |
| | Activation | ReLU |
| | FC3 | Linear(input: 64, output: 32) |
| | Output | Linear(input: 32, output: Output_dim) |
| split-CIFAR10/ split-CIFAR100 (ConvNet) | Conv1 | Conv2d(input: $C$, output: 16, kernel: $5 \times 5$) |
| | Activation | ReLU |
| | Pooling | MaxPool2d(kernel: $2 \times 2$) |
| | Conv2 | Conv2d(input: 16, output: 32, kernel: $5 \times 5$) |
| | Activation | ReLU |
| | Pooling | MaxPool2d(kernel: $2 \times 2$) |
| | Flatten | - |
| | FC1 | Linear(input: 800, output: 120) |
| | FC2 | Linear(input: 120, output: 84) |
| | FC3 | Linear(input: 84, output: Output_dim) |
| split-CIFAR10/ split-CIFAR100 (ViT) | Patch Embedding | Conv2d(input: 3, output: 192, kernel: $16 \times 16$, stride: 16) |
| | 12 Transformer Blocks | LayerNorm(shape: 192, eps: $1e{-}6$) 
 Attention: 
   Linear(input: 192, output: 576) for qkv 
   Linear(input: 192, output: 192) for projection 
 MLP: 
   LayerNorm(shape: 192, eps: $1e{-}6$) 
   Linear(input: 192, output: 768) 
   Activation: GELU 
   Linear(input: 768, output: 192) |
| | Post-Norm Classification Head | LayerNorm(shape: 192, eps: $1e{-}6$) 
 Linear(input: 192, output: Output_dim) |

