# OpenReview forum: "From Low Intrinsic Dimensionality to Non-Vacuous Generalization Bounds in Deep Multi-Task Learning"
_ICLR.cc/2026/Conference — Submitted to ICLR 2026_

### Official Review · Reviewer_kX6A · 2025-10-31

**Soundness:** 3
**Presentation:** 3
**Contribution:** 3
**Rating:** 6
**Confidence:** 3

**Summary:**

This paper studies the generalization performance of deep multi-task learning (MTL) through the lens of low intrinsic dimensionality.
It introduces the concept of Amortized Intrinsic Dimension (AID) (the effective number of parameters per task when shared representations are learned), and derives the first non-vacuous generalization bounds for deep MTL using compression and PAC-Bayesian analysis.
Empirical studies on multiple datasets show that AID is significantly smaller than the single-task intrinsic dimension, and the resulting generalization bounds are numerically non-vacuous.

**Strengths:**

1. The paper extends compression-based and PAC-Bayesian generalization analysis to multi-task learning. Two new encoding-based generalization bounds for multi-task learning that result in non-vacuous guarantees on the multi-task risk for several standard datasets and model architectures are established.

2. Experiments across several benchmarks convincingly demonstrate large reductions in intrinsic dimensionality and non-vacuous bounds.

**Weaknesses:**

1. Although the paper conceptually mentions mutual information as a measure of task relatedness, it remains largely qualitative.
There is no empirical estimation of task similarity or dependence, nor a quantitative analysis connecting such measures to the observed reductions in AID or to the resulting generalization gap.

2. Most datasets are relatively small (e.g., MNIST, CIFAR), and the models are relatively shallow compared to modern large-scale MTL or LLM settings.

3. The result is positioned as the first non-vacuous generalization bound for deep MTL, while the proof itself is relatively straightforward and closely follows prior single-task compression/PAC-Bayes analyses, so the technical novelty appears limited.

**Questions:**

1. Are there existing works that explain why low-dimensional shared representations emerge in multi-task learning, and that quantify how mutual information between task distributions influences the compression ratio or the tightness of generalization bounds?

2. Could the authors clarify what specific theoretical or technical challenges arise in this extension and how their analysis addresses them?

3. The paper fixes the threshold parameter $τ = 90$ when computing the intrinsic or amortized intrinsic dimension, yet it does not discuss how sensitive the results are to this choice. Would smaller or larger thresholds (e.g., $τ = 80$ or $95$) substantially change the estimated AID values or the resulting generalization bounds? A brief sensitivity analysis or justification for $τ = 90$ would strengthen the empirical credibility of the results.

4. Minor comment: There is a small grammatical issue in Appendix, “the Theorem 2/3” should be written as “Theorem 2/3”.

---

> ### Author Response · Authors · 2025-11-21
>
> Thank you for your review. In the following, we hope to answer your questions.
>
> > the measure of task relatedness, remains largely qualitative.
>
> Note that tightness of the bound alone is not a desirable goal. Generalization bounds can be made tighter (i.e. the gap between bound and test error reduced) by making the hypothesis class very small (e.g. just a single function), but this would result in a high test error (which the bound will correctly predict) because little or even no learning takes place anymore.
>
> Better would be a criterion for task-relatedness that guarantees a low test error itself. In a way, we do provide such a characterization, namely that tasks allow for a smaller complexity term while still learning successfully, if their AID is lower. This is not a classical distribution distance measure, of course, and it requires repeated classifier training. Indeed, we are not aware that any prior work was able to identify a criterion for task-relatedness with provable guarantees that would not require some form of model training. If you have specific examples in mind, we’ll be happy to see how the AID relates to them. It might not be possible, though: generalization bounds balance an empirical loss term with a complexity term. Controlling only the complexity term, one might lose control of the empirical loss, and estimating the empirical loss is essentially equivalent to classifier training.
>
> > Most datasets are relatively small (e.g., MNIST, CIFAR), and the models are relatively shallow compared to modern large-scale MTL or LLM settings.
>
> We respectfully disagree. Indeed, many of our experiments were chosen to match prior work from the literature on MTL with theoretical guarantees to allow for fair comparison. However, we also report results for other data domains (folktables and products), and the ViTs are a larger architecture than what prior work was able to handle. We are not aware of public benchmarks for MTL with LLMs. Is there a specific setting you had in mind?
>
> > Contributions and challenges
>
> We believe that this assessment underestimates our contribution. Indeed, in hindsight everything is obvious, but achieving the results required a combination of multiple novel contributions. Indeed, we provide easily accessible proofs for Theorem 2 and 3. However, this is a contribution in itself, as in the previous (single-task encoder-based) work, the proofs were quite complex, going through PAC-Bayesian inequalities and Kolmogorov complexity arguments. The form of the complexity term in our bounds (the length of encoding all tasks jointly) looks simple, but was important to get right, as it allows us to minimize the complexity terms much further than an almost identically looking bound would that contained the sum of individual encoding lengths. Conceptually, the main challenge, which none of the prior works had achieved, was to design a parameterization that captures the effect of information sharing in MTL, and which would achieve non-vacuous bounds. Again, we consider the fact that we succeeded to do so in simple and intuitive terms a contribution in itself.
>
> > Are there existing works that explain why low-dimensional shared representations emerge in multi-task learning, and that quantify how mutual information between task distributions influences the compression ratio or the tightness of generalization bounds?
>
> These are interesting questions, but we are not aware of any such works.
>
> > sensitivity of the results to $\tau$
>
> We used $\tau=90$ to be compatible with prior work (Li et al, 2018), where the intrinsic dimensionality for single-task learning. Other values would change the measured dimensionality, but we do not expect it to have any impact on our observation that MTL allows for much smaller dimensionality than single task learning.
>
> > Minor comment.
>
> Thanks, we fixed this in the revision.

---

> > ### Author Response · Authors · 2025-11-28
> >
> > Dear Reviewer kX6A,
> >
> > As the discussion period draws to its end, we would greatly appreciate the opportunity to engage with you regarding our rebuttal.
> >
> > We hope that our clarifications have addressed your concerns.
> >
> > Best,
> >
> > The authors

---

### Official Review · Reviewer_BXs3 · 2025-11-01

**Soundness:** 2
**Presentation:** 2
**Contribution:** 3
**Rating:** 4
**Confidence:** 3

**Summary:**

The submission derives two generalization theorems that yield the first non-vacuous numerical generalization bounds for deep multi-task learning (MTL).

The submission suggests for any deep learning architecture an alternative lower-dimensional parametrization [Eq. (1) for STL from literature, Eq. (2-3) for MTL, Eq. (7) for TL]. If I understood it correctly, the submission derives computable non-vacuous generalization bounds for models that were trained via these lower-dimensional parametrizations, by using certain compression techniques (such as quantization) for these low-dimensional parameterizations.

The submission conducts experiments for these low-dimensional parametrizations and computes non-vacuous generalization bounds. Do I understand correctly that no generalization bound was explicitly numerically computed for fully trained models in this submission?

To be honest, there are some aspects that I did not fully understand (mainly regarding what the overall storyline is). My main confusion is about two sentences on page 2:

“The bounds depend only on quantities that are available at training time and can therefore be evaluated numerically. ” and

“This view, in particular, allows us to avoid apriori parametric assumptions”

To me, these two statements seem somewhat disconnected. It appears that the paper provides (i) numerically computable bounds under specific a priori parametric assumptions (the proposed low-dimensional parametrizations), and separately (ii) more general theoretical bounds that avoid such assumptions but remain mainly conceptual without a clear numerical instantiation strategy. If I understand it correctly, (i) is kind of an example of applying (ii), but under a specific parametric assumption. So there is quite some uncertainty in my scores, and my evaluations might change based on your answers to my questions.

**Strengths:**

Multi-task Learning, Transfer Learning, and out-of-sample generalization are very important topics, and a better theoretical understanding of those is highly desirable.

Most generalization bounds in the literature are vacuous and not useful in practice. Non-vacuous generalization bounds are therefore appreciated.

While most classical learning theory focuses on Single Task Learning (STL), much of the most exciting progress in applied Deep Learning (DL) is connected to Transfer Learning (TL) and Multi-task Learning. More theory in this direction is important.

Explicitly computing non-vacuous generalization bounds on real-world datasets is a valuable contribution to the field.

**Weaknesses:**

The paper’s framing sometimes suggests that the bounds apply to arbitrary deep MTL models, whereas the actual, practically computable results seem to hold specifically for models trained under the proposed low-dimensional parametrizations (Eqs. (1), (2–3), (7)). Clarifying this scope early (perhaps even in the abstract) would prevent misunderstandings. In the current version, I got the feeling that the initial claims oversell the actual results a bit. I think the paper could be improved by communicating earlier, more clearly what the scope, the main results, and the contributions are.

Why don’t you additionally show the actual (training and) test loss in Tables 2 and 3?

Do your generalization bounds correlate with actual generalization? When you train all your low-dimensional parametrizations for different values of $k$ and $l$ (and varying other hyperparameters), I would be very interested to see the training error, the generalization error bound, and the actual generalization error for all these combinations. Computing these numbers basically comes for free for you, given that you have already trained all of them.

**Questions:**

Q1: Do I correctly understand the intrinsic dimensions of Sections 2 and 3: The definition does not give the intrinsic dimension of a fully trained model f_theta, but the intrinsic dimension only depends on the architecture, the data distribution, the training data, and an accuracy level $\tilde{A}:=\tau\%\alpha$?

I.e.,  the fully trained model f_theta is conditionally independent of the intrinsic dimension, conditioned on the architecture, the data distribution, the training data, and an accuracy level $\tilde{A}$,.

For example a model $f_\theta$ that learns very high complexity parameters, that totally overfits the training data, and expresses a very complicated high complexity function with poor generalization properties which cannot be compressed, would result in low intrinsic dimension in your experiments: As it only achieves a low test accuracy, it will be to find a low dimensional matrix P which achieves a similar test accuracy.

On the other hand, a model, $f_\tilde{\theta}$ that perfectly compressed the data in a sophisticated way (e.g. is extremely sparse), and therefore achieves a great test accuracy, would therefore correspond to a higher intrinsic dimension, as it is hard to find a low-dimensional matrix P which can match this test accuracy?

So, in other words, if I have different models $f_\theta$ with the same architecture for a given dataset, their intrinsic dimension would only vary based on their test accuracy. In a way that better test accuracy corresponds to a higher intrinsic dimension.

I think it would be good to emphasize very clearly that this notion of intrinsic dimension should not be seen as a property of the model $f_\theta$, but this notion of intrinsic dimension only depends on the architecture, the data distribution, the training data, and an accuracy level.

To me, the formulation of Definition 1 feels quite misleading: it starts with models $f_i$, but then it uses them only to compute A (which uses the test dataset), but then you don’t even directly use $A$, but only  $\tilde{A}:=\tau\%\alpha$.

This formulation could mislead a naive reader into thinking that you are computing “the intrinsic dimension of an arbitrarily trained model $f_\theta$” (in the spirit of the Local Learning Coefficient (LLC) in Singular Learning Theory) which could mislead a naive reader into thinking that those values of $\theta$ with lower intrinsic dimension correspond to good generalization (as for the LLC). However, if I understood the submission correctly, this is absolutely not what’s happening here.

I would explicitly formulate Def 1 as a property of the combination of an architecture, a data distribution, a training data set, and an accuracy level $\tilde{A}$. I don’t see any reason to introduce $\tau$, nor $A$, nor any fully trained model $f_\theta$.

Q2: My impression is that you derive generalization bounds for the parametrizations described in Eq. (1), (2-3) and (7))? But not for differently trained models? Technically, Theorems 2 and 3 are very general and could also apply to differently trained models. However, you don’t provide any explicit algorithm on how to numerically compute non-vacuous generalization bounds for fully trained DL models? Do I understand this correctly? Because your explicit compression algorithm assumes that the parameters of the model have the structure form Eq. (1), (2-3), or (7)? I think the paper would profit a lot from being crystal clear about these central questions.

Q3: Line 144: This depends on the random realization of $\theta_0$ and $P$. Do you average over multiple such realizations, or do you keep one fixed realization? Have you done some sensitivity analysis, if you get similar results for a different random seed, with a different random realization of $\theta_0$ and $P$? Do the empirical results vary a lot from seed to seed, or are they quite stable? From a mathematical point of view, do you see the ID and AID as random variables that depend on the random variables $\theta_0$ and $P$?

Q4: Do you resample $\theta_0$ for every $(k,l)$-gridpoint? Or do you sample $\theta_0$ once and keep it fixed across the grid? Would this change the results? Intuitively, I would guess that the exact random realization of $\ theta_0$ and $P$ does not matter a lot for sufficiently large models, but might matter quite a bit for smaller architectures.

Q5: Your current version of Def 1 can only be computed with access to the validation/test set in order to compute $A$?

Q6: minor remark: Line 146; “(2)/(3)” is an unusual formatting of equation numbers. “(2)-(3)”, “(2-3)”, “(2) and (3)” are more common choices, I think.

Q7: Table 1: Why does the number of tasks $n$ depend on the architecture? For example, for split-CIFAR10, two different architectures ConvNet and ViT have different values of $n$ and $m$ while the dataset stays the same.

Q8: Is $\theta_0$ random for the ViT, or do you use $\theta_0$ from the pretrained ViT? I would guess that using the pretrained $\theta_0$ results in a significantly smaller intrinsic dimension than using the random $\theta_0$. Do you agree? Have you tried that?

Q9: Why is Table 2 not referenced in the text?

Q10: Why don’t you show the actual train and test error in Tables 2 and 3? I would be very interested to see the training error, the numerically computed theoretical test error bound, and the actual test error on the test dataset for all considered combinations of $k$ and $l$.

Q11: Are the models $f_1,\dots,f_n$ in Section 5.3 trained via the parametrization from Eq. (2-3) or fully trained?

Q 12: Are the models reported in Table 3 trained via the parametrization from Eq. (7) or fully trained?

Q13: Lines 79-81: To me the sentence: “This view, in particular, allows us to avoid a priori parametric assumptions, such as similar model parameters (Evgeniou & Pontil, 2004) or the existence of a common feature space (Caruana, 1997).” feels quite confusing to me, as it seems to me that all the quantities that you can actually compute in practice from your submission also assume very specific non-standard parametrizations, i.e., Eq. (1) for STL, Eq. (2-3) for MTL, Eq. (7) for TL. Did you write this sentence because Theorems 1-3 are more generic? Do you see Theorems 2-3 as your main contribution, or Eq. (2-3) for MTL, Eq. (7) for TL?

My final score can still change in both directions depending on your answers.

---

> ### Author Response · Authors · 2025-11-21
>
> Thank you for your review. In the following, we hope to answer your questions.
>
> > computable results seem to hold specifically for models trained under the proposed low-dimensional parametrizations
>
> This is correct. We did not intend to overstate our contribution and will be happy to adjust our presentation accordingly, if you found it misleading.
>
> Our bounds (Theorem 2 and 3) do indeed hold for arbitrary MTL setups, not just for the low-dimensional representation (3). The non-vacuous numeric values we report are only for our setup. For other setups, including e.g. a shared backbone with individual classification heads, the encoding would also not be hard to do in practice: all one has to do is to store shared parts and individual parts in some (compressed) form and measure the number of bits. However, the number of parameters would be higher and the bounds likely be much larger than in our setup.
>
> We adjusted the contribution section to be clear about this.
>
> > training and test loss in Tables 2 and 3
>
> We agree that this is a good addition, we added it in the revision. The test error is generally close to the training error, within a few percent.  Additionally, you can see that the test error achieved by multi-task learning is smaller than in single-task learning.
>
> > correlation between the bounds and with actual generalization
>
> This is an interesting suggestion. We had not tried this so far, because usually generalization bounds are known not to correlate very well with test error (see e.g. Jiang et al, arXiv:1912.02178). However, as you mention, we had the values computed anyway, so we tried it, and we found reasonable correlations, see the new Figure 2 in section 5.5 of the uploaded revision.  There is a caveat, though, that we observe only part of the common bias-variance trade-off curve here, because we had only explored $(k,l)$ combinations that lead to reasonably small complexity terms. For very large values, the bound value will eventually increase, because the complexity term grows without bound. However, we suspect the test loss would likely stay low, as is commonly the case for well-trained deep networks.
>
> We will prepare a more complete plot for the final version of the manuscript.
>
> > Q1: [...] I  would explicitly formulate Def 1 as a property of the combination of an architecture, a data distribution, a training data set, and an accuracy level $\tilde{A}$. I don’t see any reason to introduce $\tau$, nor $A$, nor any fully trained model $f_\theta$.
>
> Thank you for the suggestion and we see your point. As you correctly observe, the AID is not meant to be a property of $f_1,\dots,f_n$ (which Definition 1 only uses to establish the accuracy level), but predominantly of the tasks’ data distributions, with fixing the architecture required to instantiate the parameterization and the training/validation data serving to talk about training models and estimating their accuracy). The accuracy level is essentially a free parameter.
>
> Based on your suggestion, we rephrased our definition (see revision): for the AID, we use the form you suggest with arbitrary target accuracy, so no reference models or $\tau$ factor occurs. Later, we introduce the procedure for automatically determining the actual target levels. This then exploits well-trained reference models and $\tau=0.9$, in line with (Li et al, 2018), where the intrinsic dimension of single-task learning was studied.
>
> > Q2: My impression is that you derive generalization bounds for the parametrizations described in Eq. (1), (2-3) and (7))? But not for differently trained models?
>
> You are correct. Theorem 2 and 3 are applicable to any MTL scenario and any trained models, but we only compute numeric values for them for the random subspace parametrizations. Because they are designed to have few parameters, they allow for non-vacuous results.
> For other MTL setups, such as a shared backbone with per-task classification head (fully-trained), encodings would not be hard to construct, but they have so many more parameters that the bounds will end up vacuous. This is presumably why prior MTL works showed theoretical bounds but did not report numeric values for real tasks.
> We did not do so, because our purpose is not to claim that such MTL setups are bad in any way. Our goal was to highlight the fact that standard MTL tasks do allow for low-dimensional representations, and that these can be exploited to achieve non-vacuous bounds. We will make this clear in the revision.

---

> > ### Author Response · Authors · 2025-11-21
> >
> > > Line 144 depends on the random realization of $\theta_0$ and $P$.
> >
> > The reported results are for fixed values of $\theta_0$ and $P$. However, based on our many experiments (and in line with deep learning in general), the results are quite robust and stable w.r.t. the randomness, and the results would not change noticeably for different values. Mathematically, we could define ID and AID as their expected values over the randomness of $\theta_0$ and $P$ to achieve fully deterministic definitions. So far, we did not do so to avoid further complexity in the exposition.
> >
> > > Do you resample $\theta_0$ for every $(k,l)$-gridpoint?
> >
> > The initializations are fixed, in the sense that they are generated with the same random seed. In our experiments, even for the small models the results are robust.
> >
> > > Your current version of Def 1 can only be computed with access to the validation/test set in order to compute $A$?
> >
> > Yes, we split the available data into training and validation to estimate the reference and model accuracies. The reason is that we care about learning the task, including generalization, not only about matching the training data. Following your suggestion above, we moved this aspect out of the definition of the intrinsic dimensionality, into a separate part on how to choose the reference accuracy.
> >
> > > unusual formatting of equation numbers.
> >
> > Thanks, we fixed this.
> >
> > > Table 1: different number of tasks for split-CIFAR10 ConvNet and ViT.
> >
> > The only reason is computational limitations. Because ViTs are quite bigger than ConvNets, we used a different split of the data. However the same trend is observable for both settings.
> >
> > > The pretrained $\theta_0$ for ViT experiments
> >
> > Indeed, we use the pretrained weights as $\theta_0$ in the ViT experiments. Training a ViT from scratch is not easy and requires much more data in our experiments (see line 193). Using the pretrained initialization indeed reduces the intrinsic dimension as it transfers some valuable information from ImageNet and we require to encode less information in our learning process. We would like to stress that the comparison is still fair, though, as the same information is available to the single-task as well as the multi-task learners.
> >
> > > Why is Table 2 not referenced in the text?
> >
> > Thanks for spotting this. We added it in the revised manuscript in Section 5.3, where we are talking about computing these results.
> >
> > > Why don’t you show the actual train and test error in Tables 2 and 3? I would be very interested to see the training error, the numerically computed theoretical test error bound, and the actual test error on the test dataset for all considered combinations of $k$ and $l$.
> >
> > Thank you for the suggestion. We have added the training errors (which previously were in the Appendix), and added the test errors of the selected setups. Indeed, the actual generalization gaps are actually small with differences of approximately 1 to 2 percent in the different settings.
> > We also added Figure 2, which includes the values for different choices of l and k as discussed above.
> >
> > > Are the models $f_1,\dots,f_n$ in Section 5.3 trained via the parametrization from Eq. (2-3) or fully trained?
> >
> > They are trained using our parameterization, since we want to compute the bounds for them, which includes the complexity terms. For fully-trained models, we could use some other encoders, not subspace-based, but this would incur a huge complexity cost, and the bounds would be vacuous (as they were in prior works).
> >
> > > Are the models reported in Table 3 trained via the parametrization from Eq. (7) or fully trained?
> >
> > Similar to above, we use Eq. (7), because we want to be able to evaluate their generalization bounds.
> >
> > > Lines 79-81: the quantities that you can actually compute in practice from your submission also assume very specific non-standard parametrizations.
> >
> > Indeed, we also parametrize a form in which sharing happens, but we would argue that our assumptions are less restrictive. In particular, our parametrization does not impose that all tasks have *similar* parameter vectors, or that they even coincide up to a certain network layer. Such structures can happen in the random subspace model, but they are not fixed a priori.
> > Also, our setup includes a mechanism to automatically select a subspace dimension, so anything between complete sharing and no sharing at all can emerge, depending on the task data.
> >
> >  > Do you see Theorems 2-3 as your main contribution, or Eq. (2-3) for MTL, Eq. (7) for TL?
> >
> > Both, and we find this an important point we would like to stress. We consider our contributions to be conceptual (intrinsic dimensionality for MTL), theoretical (encoding-based generalization bounds for MTL), as well as empirical (measurement of intrinsic dimensionality and non-vacuous bound values). We believe this to be a strength of our work that it spans the whole breadth of insights, rather than, e.g., just proving a new theorem or just reporting on experiments.

---

> > > ### Author Response · Authors · 2025-11-28
> > >
> > > Dear Reviewer BXs3,
> > >
> > > As the discussion period draws to its end, we would greatly appreciate the opportunity to engage with you regarding our rebuttal.
> > >
> > > We hope that our clarifications have addressed your concerns.
> > >
> > > Best,
> > >
> > > The authors

---

### Official Review · Reviewer_Rg7L · 2025-11-02

**Soundness:** 3
**Presentation:** 2
**Contribution:** 2
**Rating:** 4
**Confidence:** 4

**Summary:**

The paper studies multi-task learning and considers a setting where several tasks can be represented through a shared low-dimensional structure together with small task-specific parameters. The main idea is to describe all tasks jointly using a compact encoder and then apply a compression-style generalization bound so that the generalization error depends on the size of this joint description rather than on the sum of per-task model sizes. The authors argue that this captures the intuition that related tasks should be "paid for" only once at the level of the shared representation.

To support those claims, the authors apply a known generalization bound to this setting, introduce a notion of complexity that aims to reflect the intuition above ("amortized intrinsic dimension"), and provide empirical evidence.

Typos
- In table 1, "m" is still undefined (it's only defined later, eg in Figure 1).
- In definition 4, extra "its" in "whose its base set".
- Line 351, a "s" is missing in "Theorem 2 become ...".
- A point is missing in: the hypothesis box, Equation 8, Equation 11, Equation 14, Equation 17, Equation 18, and Equation 22.

**Strengths:**

- The discussion seems to address natural questions that one would have when reading the results (i.e., what happens in corner cases when the tasks are very related or completely different), which I appreciate.
- Experiments in Table 1 seem interesting and support their hypothesis.

**Weaknesses:**

- The statements are close to standard compression arguments ("if you have an encoder, then the total complexity is the size of the encoder + that of the encoding of the joint tasks") and seem somewhat straightforward given Theorem 1 from Shalev-Shwartz & Ben-David. The authors argue in lines 399-403 that the advantage is that the bound only depends on the encoding on the tasks all together and not the sum of the individual encodings. But in all the examples given (e.g., line 334 or line 350), it seems it is actually the sum of the individual encodings. Did I miss something? Is there an example where the joint code is shorter or are we just rewriting the sum?
- In Section 2, line 078, the authors argue "in this work, we rely on the concept that two tasks are related, if the mutual information between their data distributions is high", but such information is neither defined nor used throughout the paper. The only vague mention of it I could find is in Appendix A.4, where the authors write "Given the conceptual connections between compressibility, information, and entropy, the difference between these two quantities can be seen as a computable approximation to the mutual information between the tasks" and point to a book on Kolmogorov complexity. As currently written, this angle is more confusing than helpful; if it helps, make the link more precise, otherwise remove it.
- The accuracy used in definition 1 is the uniform average over tasks, regardless of their difficulty. It seems like it would be important to at least comment on what happens when tasks differ in difficulty or size (although I would expect the latter to be a fairly easy extension).

**Questions:**

- Could you also report the test error in Table 3? It seems like it would be interesting to have?
- In Table 2, the authors displayed the bounds of Theorems 2 and 3 side by side, as if they were comparable, and argue "the fast-rate bound offers improved guarantees compared to the more elementary Theorem 2". How so? One bounds the difference between the population and empirical risks, while the other bounds the KL. Why is it sensible to compare them?
- In Table 1, the resulting pairs $(l, k)$ are always such that $l > k$, which, if I understand correctly, means that the tasks are (sometimes very) related. Have you tried experimenting with the reverse scenario?
- Maybe a stupid question, but if I were to think about a naive approach building upon prior work discussed in Section 2, I would try to concatenate the datasets, train a single model with the same low-rank parametrization of Equation 1, and compute the intrinsic dimensionality for this "aggregated" task. If this is (much) smaller than $\text{number of tasks} \times \text{ID of a single task}$, then it would suggest it is beneficial to share the representations between the tasks. Why is this wrong?

P.S.: confidence-wise, I am somewhere between 3 and 4; I checked the paper carefully, and I have a theoretical background (not specifically in generalization bounds, although I am familiar with these), but I am not an expert in the multi-task learning literature specifically.

---

> ### Author Response · Authors · 2025-11-21
>
> Thank you for your review. In the following, we hope to answer your questions.
>
> > Typos
>
> Thank you for noticing these. We have fixed them in the revision.
>
> > The statements are close to standard compression arguments, and the advantage of encoding all tasks together.
>
> There are three steps in which the joint encoding becomes shorter than the sum of individual encodings:
> - the meta-encoder can absorb all shared parameters (each such parameter setting simply becomes a different encoder), so those are only encoded once. For our setup, that is the $\alpha$ coefficients. For a setup with shared backbone plus individual heads, it would be all parameters of the backbone, etc. This allows savings proportional to the number of tasks: $d_1 + n d_2$ instead of $n(d_1+d_2)$ if $d_1$ is the number of common parameters and $d_2$ is the number of task-specific parameters.
> - the parameter values are not stored in raw format but as indices into a learned codebook. Here, the same effect holds as in 1): the codebook becomes part of the meta-specific encoder, so it only has to be stored once instead of $n$ times.
> - the joint multi-task encoding can provide further savings: e.g. arithmetic encoding works by estimating the frequency of values and assigning codes with respect to them. For small payloads (e.g. just a few parameters of a single model), parameter values rarely repeat, the frequency estimate is very coarse, and the encoding is far from optimal. For larger payloads (the  individual parameters of all models combined), the estimates become much better and the compression comes close to its theoretical optimum.
>
> In the submission, we provide a numeric example of 2) and 3) in the appendix (Section A.4), encoding the 60 five-dimensional parameter vectors separately naively would require 60*5*32= 9600 bits. With a shared 10-value codebook and per-task arithmetic encoding, this is reduced to 2591 bits. With the same codebook and joint encoding, it is only 1124 bits in total.
>
> > task-relatedness and mutual information
>
> Thank you for the comment. Indeed, the mutual information criterion is not a formal identity, but based on the intuition that (multi-variable) mutual information relates the entropy of the joint distribution to the sum of entropies of the individual distributions. Since entropy relates to  compressibility and Kolmogorov complexity, we find it in analogy with joint compression of tasks versus individual compression.
>
> However, because mutual information is not actually measurable, and entropy only provides a lower limit on compressibility, the relation is neither an upper nor a lower bound. To avoid confusion, we removed the comment from the revision.
>
> > The accuracy used in definition 1 is the uniform average over tasks, regardless of their difficulty. It seems like it would be important to at least comment on what happens when tasks differ in difficulty or size (although I would expect the latter to be a fairly easy extension).
>
> The common convention in MTL is that even if tasks have different difficulties or number of samples, one would optimize the uniform average of per-task losses, as this expresses that all tasks are of equal importance.
> Experimentally, we would expect to still observe the same trends in AIDs vs IDs. In terms of theory, for Theorem 2 a minor difference would occur, replacing $m$ by the harmonic average of the individual sample sizes.

---

> > ### Author Response · Authors · 2025-11-21
> >
> > > test error in Table 3
> >
> > We added it in the revision in Tables 3. The test error is indeed small, typically just a few percent above the training error.  Additionally, you can see that the test error achieved by MTL +  transfer is smaller than in the no transfer setting.
> >
> > > Comparison of standard-rate and fast-rate bounds
> >
> > This is a misunderstanding. The reported values are comparable, because all of them are upper bounds on the test error (not, e.g., just the right hand side of the theorems). For Theorem 2, they are obtained by adding the empirical error to the right-hand-side complexity term. For Theorem 3, we use the fact that the left-hand-side kl-divergence is invertible (see Corollary 4 in Appendix A), and we invert it numerically by a binary search. We describe the procedure in the appendix, because it is not specific to our method, but we will be happy to expand the discussion.
> >
> > > the resulting pairs (l, k) are always such that (l > k), which means that the tasks are (sometimes very) related.
> >
> > That is an interesting observation. In all our experiments, (l,k) are determined automatically, so $l>k$ emerged by itself. We expect that $l$ would indeed end up small if tasks are truly unrelated, but we have not performed such experiments so far, since we tried to follow prior setups from the literature. If you have suggestions for data, we’ll be happy to try.
> >
> > > concatenating the datasets, and training a single model
> >
> > In multi-task learning, it is possible for the (ground-truth) target functions of different tasks to be incompatible with each other (one person likes different kinds of movies than another person), so training a single model might not reach acceptable quality, so the common goal in MTL is to find task-specific models, therefore training a single model is not desirable here.

---

> > > ### Author Response · Authors · 2025-11-28
> > >
> > > Dear Reviewer Rg7L,
> > >
> > > As the discussion period draws to its end, we would greatly appreciate the opportunity to engage with you regarding our rebuttal.
> > >
> > > We hope that our clarifications have addressed your concerns.
> > >
> > > Best,
> > >
> > > The authors

---

### Official Review · Reviewer_bsC4 · 2025-11-03

**Soundness:** 3
**Presentation:** 3
**Contribution:** 2
**Rating:** 4
**Confidence:** 3

**Summary:**

This work studies how multi-task learning can reduce the number of parameters needed to reach high accuracy by sharing structure acrosss tasks. The authors propose the concept of amortized intrinsic dimensionality, which measure the effective number of parameters each task needs when learned jointly with other tasks. The idea is that if tasks are related, they can reuse most of the representational space, so the per-task parameter requirement drops well below that of training them independently. The authors implement this through a shared low-rank parameterization: a small set of shared components captures global structure, and each task learns a few coefficients to adapt them. They empirically show across diverse datasets that as the number of tasks increases, tha amortized parameter cost decreases while the accuracy stays mostly constant. This provides evidence that multi-task learning discovers a shared low-dimensional subspace that explains why such models compress and generalize well, resulting in better generalization bounds.

**Strengths:**

- The authors clearly demonstrate that the multi-task setting results in higher compressibility and better bounds "in their particular setting". The rigorous formulation makes the multi-task efficiency measurable and comparable to the single task setting.
- The amortized intrinsic dimensionality provides a simple defitition of 'how much sharing happens' and can be evaluated directly from experiments.
- There is a clear scaling behavior as the number of tasks grows, where the amortized dimension drops continuouslt, which supports the central claim that related tasks occupy a common representation space.

**Weaknesses:**

- The paper’s contribution is mostly conceptual and empirical as it does not show theoretically that if the tasks are related, e.g., measured in some form of distributional distance, the bounds will be tighter compared to the single task setting. It would be great to provide a sufficiency theoretical guarantee where if the task are related given a certain metric, the bounds will be theoretically improved.
- There is no analysis to determine a priori whether the tasks are related or to identify negative transfer when unrelated tasks are included. The experimental setup is limited to very simple and clearly related tasks.
- While results are support the main claim that related rasks result in better bound, the paper doesn’t offer prescriptive recipes for how to choose related tasks or how many to choose in order to improve generalization; back to the distance measurement question.
- The experiments could be extended to more complicated datasets that are harder to perform well on, where the multi-task setting would have clearer empirical benefits to be tested against the bounds.
- The bounds are not compared against sota techniques to obtain tight bounds: what if sota techniques for single task result in better bounds than the multi-task bounds?

**Questions:**

- Can you provide a theoretical sufficiency guarantee: for example, proving that if tasks are related under a defined distance metric, the amortized bound will be tighter than the single-tasks one?
- Is there a quantitative criterion for task relatedness that could predict when multi tak will help or harm performance? Could mutual information or divergence measures serve that role?
- How does the framework work in the presence of unrelated tasks? Can you empirically test for this case?
- Can you suggest a practical recipe for selecting related tasks or estimating the right number of tasks to group jointly to improve generalization?
- Would extending experiments to harder datasets or less correlated task suites change the observed scaling trends?
- How do your amortized bounds compare numerically to state-of-the-art single-task generalization bounds? is the improvement consistent when using the best available baselines?

---

> ### Author Response · Authors · 2025-11-21
>
> Thank you for your review. In the following, we hope to answer your questions.
>
> > The paper’s contribution is mostly conceptual and empirical
>
> Our contribution is in fact, conceptual (intrinsic dimensionality for MTL), theoretical (encoding-based generalization bounds) and empirical (measurement of intrinsic dimensionality  and non-vacuous bound values). We believe this to be a strength of our work, that it spans a breadth of insights, not just one specific aspect.
>
> > proving that if tasks are related under a defined distance metric, the amortized bound will be tighter than the single-tasks one?
>
> Note that tightness of the bound alone is not a desirable goal. Generalization bounds can be made tighter (i.e. the gap between bound and test error reduced) by making the hypothesis class very small (e.g. just a single function), but this would result in a high test error (which the bound will correctly predict) because little or even no learning takes place anymore.
>
> Better would be a criterion for task-relatedness that guarantees a low test error itself. In a way, we do provide such a characterization, namely that tasks allow for a smaller complexity term while still learning successfully, if their AID is lower. This is not a classical distribution distance measure, of course, and it requires repeated classifier training. Indeed, we are not aware that any prior work was able to identify a criterion for task-relatedness with provable guarantees that would not require some form of model training. If you have specific examples in mind, we’ll be happy to see how the AID relates to them. It might not be possible, though: generalization bounds balance an empirical loss term with a complexity term. Controlling only the complexity term, one might lose control of the empirical loss, and estimating the empirical loss is essentially equivalent to classifier training.
>
> > a quantitative criterion for task relatedness that could predict when multi task will help or harm performance
>
> The question of when multi-task learning is beneficial and when it is not is a much more fundamental one than our contributions. We are not aware of such criteria in a priori form, and they might be impossible due to no-free-lunch-type arguments.
> Note that our setup does include an implicit fall-back option from multi-task to single-task learning: combining completely unrelated tasks (no even theoretical benefit in sharing) would lead the AID growing linearly with the number of tasks (e.g. $l=n, k=1$), so there is enough capacity to learn each task separately. The bounds would also collapse to the averaged single-task setup with just minor overhead.
>
> > a practical recipe for selecting related tasks or estimating the right number of tasks to group jointly to improve generalization
>
> See above, the AID itself can be used to determine which tasks are compressible together. We are not aware of criteria of this type for other MTL setups, either, except heuristic ones.
>
> > more complicated datasets
>
> Our experiments were meant to reflect prior work with theoretical guarantees from the MTL literature. We agree that larger-scale experiments would be interesting, even if --depending on the model choice-- the resulting bounds might not be non-vacuous anymore. If you are aware of any large-scale datasets for multi-task classification with many classification tasks, we’d be grateful if you let us know. We have no reason to expect that our findings would change, though. We already test different model architecture (MLP, ConvNet, ViT) and data modalities (images, tabular, text), as the findings are completely consistent.
>
> > comparison numerically to state-of-the-art single-task generalization bounds
>
> The statement that we do not compare to SOTA is incorrect. Our single-task baseline is the bound of (Lotfi et al, 2022), which is exactly the state-of-the-art for single-task generalization bounds for deep networks.

---

> > ### Author Response · Authors · 2025-11-28
> >
> > Dear Reviewer bsC4,
> >
> > As the discussion period draws to its end, we would greatly appreciate the opportunity to engage with you regarding our rebuttal.
> >
> > We hope that our clarifications have addressed your concerns.
> >
> > Best,
> >
> > The authors

---

### Official Review · Reviewer_hMHb · 2025-11-05

**Soundness:** 3
**Presentation:** 3
**Contribution:** 2
**Rating:** 4
**Confidence:** 4

**Summary:**

The paper proposes a parameter-sharing approach for multi-task learning based on a hierarchical parameterization of an intrinsic-dimensional model. Specifically, the method represents a set of shared parameters as (k) (l)-dimensional vectors, each projected into the network parameter space of dimension (D) using a different random matrix (P_i), and models each task as a linear combination of these shared components. The authors leverage this parameter-sharing structure to apply compression-based PAC-Bayesian bounds, leading to non-vacuous generalization guarantees in multi-task learning settings. Empirically, they demonstrate that this parameter-sharing strategy can achieve a pre-set 90% accuracy while operating in a significantly lower "amortized"-intrinsic dimensionality, thereby enjoying favorable generalization guarantees.

**Strengths:**

* The paper is clearly written and well organized.
* The proposed approach is conceptually sound and well motivated.
* The computed PAC-Bayesian bounds are non-vacuous, which is a meaningful result for multi-task learning.

**Weaknesses:**

* The technical novelty appears limited.
* The work largely builds upon the compression-based PAC-Bayesian framework of Lotfi et al. (2022) and extends it in a relatively straightforward manner.
* The benefit of the proposed hierarchical parameter-sharing approach over existing methods is not clearly demonstrated.
* It remains unclear how the method compares to simpler parameter-sharing schemes, such as decomposing the parameter (w) into shared and task-specific components as in Li et al. (2018).

**Minor Comments:**

* Reporting test set errors alongside the theoretical guarantees would strengthen the empirical evaluation.
* For the Transformer-based experiments, it would be useful to show the performance of the pretrained Vision Transformer on CIFAR without fine-tuning.
* Since Lotfi et al. (2022) evaluated their compression-based PAC-Bayesian framework on larger-scale datasets such as ImageNet, it would be valuable for the authors to test their approach on a comparable scale to better assess its general applicability and scalability.

**Questions:**

See the weaknesses section.

---

> ### Author Response · Authors · 2025-11-21
>
> Thank you for your review. In the following, we hope to answer your questions.
>
> > Contributions and comparison to Lotfi et al.
>
> This seems to be a misunderstanding of our contributions:
>
> - we introduce a notion of intrinsic dimensionality for multit-task learning,
> based on a new (learnable) low-rank parametrization
> - we prove new, explicitly computable, generalization bounds for general multi-task
> - we report the first numerically non-vacuous generalization guarantees for deep MTL.
>
> We consider it a strength of our work that we cover the conceptual, the theoretical and the empirical level, not just improving one of the aspects.
>
> Part 2) indeed extends prior (single-task) work from Lotfi et al, but technically it is not as straight-forward as it might look in hindsight. First, the proof we provide is much easier, as it avoids advanced concepts such as PAC-Bayes inequalities and Kolmogorov complexity. We consider this a contribution in itself, as our form will allow easier extension to other setups.
> Second, the form of the complexity term in our bounds (the length of encoding all tasks jointly) was important to get right, because it allows for clearly tighter numeric value than the canonical term which would contain the sum of the individual encoding lengths.
>
> > Comparison to other MTL methods
>
> We do not claim that our parameter-sharing approach is superior to other ways of MTL, see our discussion of contributions above. Our approach provides an intuitive approach to quantifying the intrinsic dimensionality of MTL problems, and it can be exploited to obtain non-vacious generalization guarantees, which have not been demonstrated for previous MTL techniques.
>
> > comparison to simpler parameter-sharing schemes, as in Li et al. (2018).
>
> Could you please clarify the suggestion? Li et al. (2018) is not about multi-task learning. It studies single-task learning in the random-subspace parametrization.
>
> Do you mean to split parameters into groups, like a shared backbone and task-specific classification head, as in (Caruana 1997)? Or do you suggest an additive decomposition, as in (Evgeniou et al, 2004), but inside a random subspace representation,  $\theta_i = P(w_{shared} + w_i)$? Both would be covered by our theorems, as those are not restricted to any specific parametrization. We would not expect non-vacuous bounds, though, as the number of learnable parameters would still be larger than in our setup.
>
> > Reporting test set errors
>
> We agree, we added the results in Tables 2. The test error is indeed small, typically just a few percent above the training error. Additionally, you can see that the test error achieved by multi-task learning is smaller than in single-task learning.
>
> > Vision Transformer on CIFAR without fine-tuning.
>
> Unfortunately, this is not possible. As it is standard, the ViT was trained on the ImageNet dataset, so the classification head is incompatible with CIFAR.
>
> > larger-scale datasets such as ImageNet, as in Lotfi et al. (2022)
>
> We agree that large-scale experiments would be interesting. Mainly, our experiments were meant to allow a fair comparison to prior work on multi-task learning with generalization guarantees. Additionally, we wanted to include data modalities beyond images (Folktables, Products), and other architectures (ViTs). To our knowledge there is no established multi-task classification benchmark of ImageNet size, but if you have one in mind, please let us know.
>
> One caveat is that in order to be comparable to other approaches, for ImageNet one has to train ResNet architectures that include BatchNorm layers, and these need special treatment in parametrization and bounds because of their learned BatchNorm statistics.
>
> Unfortunately, Lotfi et al. (2022) forgot to account for these when evaluating their bound, so the numeric bound values they report for ImageNet are actually incorrect.

---

> ### Author Response · Authors · 2025-11-28
>
> Dear Reviewer hMHb,
>
> As the discussion period draws to its end, we would greatly appreciate the opportunity to engage with you regarding our rebuttal.
>
> We hope that our clarifications have addressed your concerns.
>
> Best,
>
> The authors

---

### Meta-Review · Area_Chair_DtJ5 · 2026-01-12

**Summary:**

This submission analyzes the generalization of multi-task learning through the lens of low intrinsic dimensionality. The reviewers raised the following concerns.

* Reviewers Rg7L, hMHb, and kX6A raised concerns that the theoretical contribution is incremental, since the analysis resembles standard compression arguments from prior work. The authors partly addressed this concern by explaining the conceptual novelty of the proposed notion of low intrinsic dimensionality for MTL.

* Reviewers bsC4, hMHb, and kX6A complained about inadequate experiments (on more complicated datasets) to demonstrate the tightness of the bounds and advantages over existing approaches. Reviewer BXs3 also asked whether the proposed generalization bound correlates with the actual test error; in the revision, the authors added a figure to examine this correlation.

* Reviewers bsC4 and kX6A commented that the conceptual contribution based on task relation is mostly qualitative and does not offer prescriptive recipes. The authors partly addressed this concern by explaining how their proposed criterion of task relatedness informs the complexity of MTL learning.

Overall, the authors' response addressed some of the concerns raised by the reviewers. However, the area chair believes that this submission would benefit from another round of revision to highlight the technical novelties and strengthen the empirical validation.

**Reviewer Concerns:**

See above.

**Reviewer Scores:**

Reviewers hMHb, bsC4, Rg7L, and BXs3 all gave a score of borderline reject. The authors addressed some of the concerns in the discussion period (especially the clarification questions from BXs3); however, the area chair believes that concerns about the lack of quantitative and empirical validation, and the limited theoretical contribution, are not fully addressed.

---

### Decision · Program_Chairs · 2026-01-26

Reject